# FISHER DIVERGENCE FOR ATTRIBUTION

## ABSTRACT

Feature attribution methods aim to explain model predictions by identifying input features that are most influential to the output. While perturbation-based methods are intuitive and widely used, they often rely on restricted or discretized perturbation spaces, limiting their ability to capture the complex dependencies in high-dimensional data. In this work, we propose a novel framework that defines the perturbation space as a continuous-time stochastic differential equation (SDE), enabling a general, unbounded formulation of the perturbation space. This formulation significantly increases the expressiveness of the perturbation space while introducing new optimization challenges. To address this, we derive a theoretical connection between the KL divergence and the Fisher divergence under general SDEs, and further establish that the time derivative of mutual information between perturbed and original inputs is governed by the Fisher divergence. These results allow us to simplify the attribution objective and compute pointwise information as feature importance scores. Empirical results on large-scale image classification tasks show that our method produces sharper, more coherent, and better localized attribution maps compared to existing approaches, as demonstrated by both qualitative visualizations and quantitative evaluations.

## 1 INTRODUCTION

Deep neural networks (DNNs) have demonstrated remarkable success in a wide range of applications, including computer vision, and natural language processing. Despite their impressive performance, DNNs often function as "black boxes," providing limited insight into their underlying decision-making processes Pan et al. (2021); Novello et al. (2022); Chen et al. (2024). This lack of transparency raises concerns in high-stakes domains where understanding the rationale behind model predictions is vital for trust, accountability, and adherence to regulatory guidelines Chaddad et al. (2023); Saraswat et al. (2022); Soundararajan & Shenbagaraman (2024). Consequently, explainability has emerged as a pivotal research area, aiming to elucidate how neural networks transform inputs into outputs Van der Velden et al. (2022); Bai et al. (2021).

One well-established strategy for enhancing neural network explainability is feature attribution, which quantifies how each input feature contributes to a model's output Zhou et al. (2022). Although various techniques exist to achieve this, perturbation-based methods—which systematically modify or remove parts of the input and then measure changes in the model's predictions—are particularly intuitive, offering a clear and direct interpretation of how the model reacts to input modifications and their influence on its behavior Ivanovs et al. (2021).

A straightforward perturbation-based approach is to consider a feature as important if a small changes of the feature leads to a significant prediction change (**Small Perturbations → Large Output Changes**). Gradient-based methods operationalize this idea by computing the gradient of the output with respect to the input Simonyan et al. (2014); Sundararajan et al. (2017); Smilkov et al. (2017). However, a major limitation is that their dependence on narrowly defined perturbation spaces Fel et al. (2023) may fail to trigger desirable changes in the function's output, thereby overlook certain aspects of feature importance.

When the perturbation space is broadened to allow larger input changes, both inputs and outputs can shift significantly (**Large Perturbations → Large Output Changes**). In such scenarios, to assess feature importance is to linearly or non-linearly distribute the total output change across individual input features based on its share of the output change. However, linear decomposition is

only suitable for small-scale perturbations. Non-linear decomposition methods like SHAP and its variants Lundberg & Lee (2017); Jethani et al. (2021); Tsai et al. (2023), while theoretically sound, are computationally expensive.

Another perspective identifies important features by alternating features that cause minimal impact on the model's output and thereby maintain important features in the input (**Large Perturbations → Small Output Changes**). However, exploring large input changes demands navigating a vast perturbation space, which is computationally expensive, especially for high-dimensional data like images. To mitigate this, many methods reduce the complexity by partitioning the inputs into smaller blocks Novello et al. (2022) or by sampling within the predefined perturbation space Fel et al. (2023) rather than performing an exhaustive search. Although these techniques lower computational overhead, they also sacrifice precision: coarse partitioning overlooks fine-grained details, and limited sampling may fail to represent the entire perturbation space, potentially leading to imprecise or misleading attributions.

To overcome the limitations of fixed-scale perturbations, we model the entire perturbation space as a continuous-time Stochastic Differential Equation (SDE). Instead of relying on a finite collection of noise distributions, we consider a smooth, time-evolving family of distributions generated by a predetermined diffusion process. To the best of our knowledge, this is the first work to formalize the perturbation space for attribution using continuous-time SDEs, offering a more general and principled modeling framework than prior discretized or fixed-noise approaches.

Although defining the perturbation space via an SDE greatly increases its expressiveness, it also makes direct search over this space intractable. To address this challenge, we leverage the mathematical properties of the SDE to innovatively derive a connection between KL divergence and Fisher divergence, and further link mutual information to the time-integrated Fisher divergence. This result simplifies the optimization objective into a tractable form, and allows us to compute mutual information as attribution scores by training a neural network to estimate the marginal score function.

Our contributions can be summarized as follows:

- **SDE for Perturbations Space:** We propose to define an unconstrained perturbation space with SDEs, rather than a predefined space of fixed size.

- **A Novel and tractable Simplified Optimization Framework for Attribution:** We propose a simplified optimization framework for feature attribution in an unconstrained perturbation space.

- **Theoretical results:** We establish a theoretical connection between KL divergence and Fisher divergence under general time-dependent SDEs. Specifically, we prove that the time derivative of the KL divergence is proportional to the Fisher divergence, and further show that the time derivative of mutual information is also governed by the Fisher divergence.

## 2 RELATED WORK

Explainable methods based on input perturbations can be broadly categorized according to how changes in inputs (small or large) correlate with changes in the model's output (small or large). Below, we discuss three main approaches: (1) small perturbations leading to large output changes, (2) large perturbations leading to large output changes, and (3) large perturbations leading to small output changes.

**Small Perturbations → Large Output Changes:** If a minor change in a particular feature leads to a substantial difference in the model's output, that feature is deemed influential. Gradient-based methods adopted this strategy via gradients. Saliency Maps Simonyan et al. (2014) compute these gradients to highlight important features. Integrated Gradients Sundararajan et al. (2017) and Smooth-Grad Smilkov et al. (2017) also utilize this concept but employ different data augmentation strategies to enhance the estimation of feature importance. Grad-CAM Selvaraju et al. (2017a) preserves model flexibility by leveraging the gradients of the target class with respect to the activations in the last convolutional layer, highlighting regions of importance. Building on this, Guided-GradCAM Selvaraju et al. (2017b) enhances Grad-CAM by integrating it with Guided-BP Springenberg et al. (2014), providing finer and more detailed insights into feature importance.

**Large Perturbations → Large Output Changes:** Various methods employ this principle of "big changes, big effects." RISE Petsiuk et al. (2018) uses multiple random binary masks to occlude large portions of the input; the change in model output across many masked samples is aggregated into a saliency map. LIME Ribeiro et al. (2016) perturbs different subsets of features and then fits a simple local model to approximate how drastically these features affect the original prediction. Meanwhile, SHAP Lundberg & Lee (2017) and its variants (FaithSHAP, FastSHAP Jethani et al. (2021); Tsai et al. (2023)) draw on the game-theoretic concept of Shapley values to systematically credit (or blame) each feature for the overall output change when large combinations of features are removed. Beyond single-pass masking, Remove and Retrain (ROAR) Hooker et al. (2019) retrains the model after removing key features, measuring performance deterioration to verify feature importance. Lastly, Explaining by Removing Covert et al. (2021) provides a unified theoretical framework for such removal-based strategies, analyzing how various large-perturbation methods align with desired explanation properties.

**Large Perturbations → Small Output Changes:** This approach focuses on identifying features that are critical to the model's prediction even when large portions of the input are altered but the output change remains minimal. For instance, studies have shown that improving network interpretability can directly enhance adversarial robustness by ensuring predictions rely on stable feature subsets Boopathy et al. (2020). Abduction-based explanations have been proposed to identify invariant features that maintain predictions under significant input perturbations Ignatiev et al. (2019), while metrics for representativity and consistency have been introduced to evaluate explanation stability in these scenarios Fel & Vigouroux (2022). Robustness-based evaluation methods further ensure that explanations remain reliable across large perturbation spaces Hsieh et al. (2021). Additionally, research has emphasized the importance of stable baselines and modeling uncertainty in feature attribution methods to ensure consistent and interpretable explanations when extensive input changes occur Sturmfels et al. (2020); Slack et al. (2021). EVA Fel et al. (2023) introduces a framework that guarantees the stability of explanations across a predefined perturbation space, ensuring that changes in unimportant features do not mislead the explanation.

## 3 PROBLEM DEFINITION

The *Perturbation Space* defines how input features are modified to explore the model's response to different input variations. For a machine learning model $f : \mathbb{R}^n \to \mathbb{R}$ that maps an input feature vector $\mathbf{x} = [x_1, x_2, \ldots, x_n]^\top$ to an output $f(\mathbf{x})$, a general perturbation of a single feature $x_i$ can be expressed as:

$$x_i' = \mu_i x_i + \delta_i, \tag{1}$$

where $\mu_i \in \mathbb{R}$ is a multiplicative scaling factor and $\delta_i \in \mathbb{R}$ is an additive noise term drawn from a distribution $D_i$. The perturbed input is then denoted by $\mathbf{x}' = [x_1', x_2', \ldots, x_n']^\top$. Varying $\mu_i$ and $\delta_i$ allows the construction of different types of perturbation strategies, each representing a specific assumption about how feature variations influence model predictions.

However, most existing attribution methods impose strong structural constraints on the perturbation space. Gradient-based techniques, such as Saliency Maps Simonyan et al. (2014), SmoothGrad Smilkov et al. (2017), and Integrated Gradients Sundararajan et al. (2017), rely on infinitesimal perturbations where $\mu_i = 1$ and $\delta_i = dx$, limiting the exploration to local linear approximations. Occlusion-based approaches like RISE Petsiuk et al. (2018) set $\mu_i = 0$, $\delta_i = 0$, effectively masking selected features. Reference-based methods such as DeepLIFT Shrikumar et al. (2017) perturb features towards a predefined baseline $b_i$ by setting $\mu_i = 0$, $\delta_i = b_i$. More flexible methods may allow perturbations with bounded norms, such as $\|\delta_i\| \in (0, L)$, where $L$ is a manually chosen hyperparameter Fel et al. (2023). While these techniques offer some level of flexibility, they are still limited to discrete or pre-defined perturbation forms, lacking the capacity to explore rich, continuous variations in input space.

To overcome these limitations, we propose to define the perturbation space using a continuous stochastic process governed by a stochastic differential equation (SDE). Unlike fixed or bounded perturbations, an SDE enables the generation of a dense family of perturbed inputs that evolve continuously from the original input. The perturbation process is defined by:

$$dX_t = \mu(t)\, dt + \sigma(t)\, dW_t, \tag{2}$$

where $X_t$ denotes the perturbed input at time $t$, $\mu(t)$ is a time-dependent drift term controlling the deterministic component, $\sigma(t)$ is a diffusion coefficient scaling the injected noise, and $dW_t$ is the standard Wiener process. By varying the perturbation time $t$, this formulation allows us to flexibly explore a wide continuum of corrupted inputs ranging from minimally to heavily perturbed, without being restricted to a specific functional form.

While our SDE-based formulation enables a continuous and expressive perturbation space, it is computationally infeasible to evaluate the model's response across all possible combinations of perturbed features. Exhaustively enumerating perturbation patterns quickly becomes intractable in high-dimensional settings. To circumvent this challenge, we adopt a search strategy based on the principle that **Large Perturbations $\rightarrow$ Large Output Changes**. That is, we seek the maximal perturbation to the input that induces minimal change in the model's output, thereby exposing the subspace of the input that is most critical to the prediction.

We formalize this as a constrained optimization problem:

$$\begin{aligned} \underset{\mathbf{x}'}{\text{maximize}} \quad & \|\mathbf{x}' - \mathbf{x}\| \\ \text{subject to} \quad & |f(\mathbf{x}') - f(\mathbf{x})| \leq \xi, \end{aligned} \tag{3}$$

where $\mathbf{x} \sim X$ is sampled from the original data distribution, $\mathbf{x}' \sim X'$ is a perturbed version drawn from the SDE-defined space, and $\xi$ is a small threshold ensuring that the model's output remains approximately constant. This objective seeks perturbed inputs that are maximally different from the original inputs while preserving prediction confidence. However, simply using the norm $\|\mathbf{x}' - \mathbf{x}\|$ as the objective function may not yield semantically meaningful results. This naive formulation ignores the informational structure of the input and treats all perturbations as equally important, potentially overlooking complex dependencies among features.

To overcome this limitation, we instead take an information-theoretic perspective and adopt the mutual information $I(X; X')$ between original and perturbed data distributions as the objective to minimize. The intuition is that, if a perturbed input $\mathbf{x}'$ retains little information about its source $\mathbf{x}$, then the removed components are likely non-essential for prediction. Formally, given an original dataset $X$ and a perturbed counterpart $X'$, our goal is to find a transformation that minimizes $I(X; X')$ while ensuring that the classifier's output distribution remains consistent. This objective aligns with the *Information Bottleneck* (IB) principle Alemi et al. (2016), which seeks a compressed representation that preserves task-relevant information.

We define the following IB-inspired loss function:

$$\mathcal{L}_{\text{IB}} = I(X; X') - I(X'; Y), \tag{4}$$

where $I(X; X')$ quantifies the redundancy between the input and its perturbation, and $I(X'; Y)$ captures the information retained for predicting the label $Y$.

Based on the above formulation, our attribution objective consists of two goals:

1. **Searching optimal $\mathbf{x}'$.** We seek to identify a distribution $X'$ within the continuous perturbation space, such that for each input $\mathbf{x} \sim X$, there exists a corresponding perturbed input $\mathbf{x}' \sim X'$ that minimizes the mutual information $I(X'; X)$ while preserving the model's output. This optimization ensures that only the most necessary features for prediction are retained in the perturbed input.

2. **Computing Attribution Score.** To compute fine-grained attributions, we evaluate the *pointwise information* between $\mathbf{x}$ and $\mathbf{x}'$ as a feature-wise importance score. For a given pair $(\mathbf{x}, \mathbf{x}')$, the property of *pointwise information* is given as (Kong et al. (2023); Levy & Goldberg (2014))

$$I(X; X') = \mathbb{E}_{p(\mathbf{x}, \mathbf{x}')} \left[ \boldsymbol{i}(\mathbf{x}; \mathbf{x}') \right], \tag{5}$$

where $\boldsymbol{i}(\mathbf{x}; \mathbf{x}')$ quantifies the contribution of each input dimension to the mutual information.

# 4 SOLUTION

## 4.1 TIME DERIVATIVE OF MUTUAL INFORMATION VIA FISHER DIVERGENCE

We generalize the seminal result of Lyu (2012), which is confined to the SDE $dY_t = dW_t$ (zero drift, unit diffusion), to a fully time-dependent yet *spatially homogeneous* setting that allows both the drift and the diffusion strength to vary with time:

$$dY_t = \mu(t)\, dt + \sigma(t)\, dW_t, \qquad t \geq 0, \tag{6}$$

where $\mu : [0, \infty) \to \mathbb{R}^d$ and $\sigma : [0, \infty) \to \mathbb{R}_{\geq 0}$. This formulation is essential in modern generative frameworks—notably score-based diffusion processes (Song et al., 2020)—where noise schedules and drift behaviors evolve dynamically over time while remaining spatially homogeneous.

In what follows we rely on the Fokker–Planck channel identities of Wibisono et al. (2017) and record their specialization to our time–dependent yet spatially homogeneous setting. For a general FP channel $\partial_t r = -\nabla \cdot (ar) + \frac{1}{2}\nabla \cdot (\nabla \cdot (br))$ with diffusion tensor $b(x, t) \succeq 0$, (Wibisono et al., 2017, Thms. 6–7) show

$$\frac{d}{dt} D_{\mathrm{KL}}(p_t \| q_t) = -\frac{1}{2} \int p_t(x) \left(\nabla \log p_t - \nabla \log q_t\right)^\top b(x, t) \left(\nabla \log p_t - \nabla \log q_t\right) dx, \tag{7}$$

$$\frac{d}{dt} I(X_t; X_0) = -\frac{1}{2} \mathbb{E}\Big[\big(s_t(X_t \mid X_0) - s_t(X_t)\big)^\top b\big(X_t, t\big) \big(s_t(X_t \mid X_0) - s_t(X_t)\big)\Big]. \tag{8}$$

(Informally, when $p_t$ and $q_t$ solve the *same* channel, the drift contributions cancel by two integrations by parts, so only diffusion dissipates KL/MI via the $b$–metric on score differences; see Wibisono et al. (2017) or Appendix A for a detailed alternative derivation in our notation).

In our setting $dY_t = \mu(t)\, dt + \sigma(t)\, dW_t$ the diffusion is spatially homogeneous, i.e., $b(x, t) \equiv \sigma(t)^2 I$. Writing the (relative) Fisher divergence

$$D_F(p \| q) := \int p(x) \, \|\nabla \log p(x) - \nabla \log q(x)\|^2 \, dx,$$

the quadratic forms in equation 7–equation 8 reduce to

$$\int p_t \, (\cdot)^\top b_t(\cdot) = \sigma(t)^2 \, D_F(p_t \| q_t), \qquad \mathbb{E}\big[(\cdot)^\top b_t(\cdot)\big] = \sigma(t)^2 \, \mathbb{E}\big[\|s_t(X_t \mid X_0) - s_t(X_t)\|^2\big].$$

Substituting into equation 7–equation 8 yields the identities we use throughout:

$$\frac{d}{dt} D_{\mathrm{KL}}(p_t \| q_t) = -\frac{\sigma(t)^2}{2} \, D_F(p_t \| q_t), \tag{9}$$

$$\frac{d}{dt} I(X_t; X_0) = -\frac{\sigma(t)^2}{2} \, \mathbb{E}\big[\|s_t(X_t \mid X_0) - s_t(X_t)\|^2\big]. \tag{10}$$

## 4.2 SEARCHING OPTIMAL $\mathbf{x}'$

From the Eq. 10, integrating along the same FP channel yields (in what follows, we omit the constant terms for convenience)

$$I(X_1; X_0) = -\frac{1}{2} \int_0^1 \sigma(t)^2 \, \mathbb{E}\big[\|s_t(X_t \mid X_0) - s_t(X_t)\|^2\big] \, dt + Const. \tag{11}$$

Following (Kong et al., 2023; Levy & Goldberg, 2014), define pointwise information:

$$\boldsymbol{i}(\mathbf{x}; \mathbf{x}') = -\frac{1}{2} \int_0^1 \sigma(t)^2 \, \mathbb{E}\Big[\|s_t(\mathbf{x}' \mid \mathbf{x}) - s_t(\mathbf{x}')\|^2\Big] \, dt. \tag{12}$$

By construction, the squared Euclidean norm of the score difference naturally decomposes coordinate–wise:

$$\|s_t(\mathbf{x}' \mid \mathbf{x}) - s_t(\mathbf{x}')\|^2 = \sum_{i=1}^d \big(s_{t,i}(\mathbf{x}' \mid \mathbf{x}) - s_{t,i}(\mathbf{x}')\big)^2.$$

Since both the expectation and the time integral are linear operators, and the input dimension $d$ is finite, Fubini–Tonelli's theorem guarantees that the order of summation, expectation, and integration can be interchanged without difficulty. Consequently, the mutual information integral decomposes exactly into a sum of per–coordinate contributions:

$$\frac{1}{2}\int_0^1 \sum_{i=1}^d \mathbb{E}\Big[\big(\Delta s_{t,i}\big)^2\Big]\, dt \;=\; \sum_{i=1}^d \; \underbrace{\left(\frac{1}{2}\int_0^1 \mathbb{E}\Big[\big(\Delta s_{t,i}\big)^2\Big]\, dt\right)}_{\text{complete contribution of pixel } i}, \tag{13}$$

where $\Delta s_{t,i} = s_{t,i}(\mathbf{x}' \mid \mathbf{x}) - s_{t,i}(\mathbf{x}')$ denotes the $i$-th coordinate of the score difference. Therefore, the total mutual information (i.e., the expectation of $\boldsymbol{i}$) is exactly the sum of each pixel's complete integral contribution over $[0,1]$.

It is also possible to assign each pixel dimension its own upper integration limit $t_i \in [0,1]$:

$$C_i(t_i) \;\triangleq\; \frac{1}{2}\int_0^{t_i} \sigma(t)^2 \mathbb{E}\Big[\big(\Delta s_{t,i}\big)^2\Big]\, dt. \tag{14}$$

Summing over all pixels gives

$$\sum_{i=1}^d C_i(t_i) \;=\; \sum_{i=1}^d \frac{1}{2}\int_0^{t_i} \sigma(t)^2 \mathbb{E}\Big[\big(\Delta s_{t,i}\big)^2\Big]\, dt. \tag{15}$$

If all $t_i = 1$, then $\sum_i C_i(t_i)$ recovers the total mutual information. If some pixels have $t_i < 1$, then the result is a lower bound of the total mutual information (partial integration), that is $\sum_i C_i(t_i) \leq \sum_i C_i(1)$. Intuitively, smaller $t$ corresponds to smaller noise (more information), so truncating the integral of a pixel at $t_i < 1$ means only accumulating its contribution from the early noisy regime. At the infinitesimal (differential) level, the Fisher divergence can still be consistently evaluated at each fixed $t$, since the spatial homogeneity assumption continues to hold.

Our objective is to minimize mutual information while preserving the model's predictive performance. Therefore, we combine $L_{\mathrm{MI}}$ with a task-specific loss function, such as the cross-entropy loss $L_{\mathrm{CE}}$ for classification tasks Zhang et al. (2021); Schulz et al. (2020):

$$L \;=\; \beta\, L_{\mathrm{MI}} \;+\; L_{\mathrm{CE}}, \qquad L_{\mathrm{MI}} \;=\; \sum_{i=1}^d C_i(t_i). \tag{16}$$

where $\beta > 0$ is a trade–off hyperparameter that controls the relative strength of the mutual–information term.

## 4.3 Implementation

We implement the mutual-information objective by leveraging a *pretrained* score-based diffusion model and a lightweight adapter that predicts pixel-specific truncation times. Concretely, let $s_\theta(\cdot, t)$ denote a frozen score network trained on the forward SDE with spatially homogeneous diffusion, and let the forward noising kernel be $q_t(\mathbf{x}_t \mid \mathbf{x}_0) = \mathcal{N}(\alpha(t)\mathbf{x}_0, \sigma(t)^2 \mathbf{I})$. For any $(\mathbf{x}_0, \mathbf{x}_t)$ pair, the conditional score is available in closed form:

$$s_t(\mathbf{x}_t \mid \mathbf{x}_0) \;=\; \nabla_{\mathbf{x}_t} \log q_t(\mathbf{x}_t \mid \mathbf{x}_0) \;=\; -\frac{\mathbf{x}_t - \alpha(t)\mathbf{x}_0}{\sigma(t)^2}.$$

We approximate the marginal score by the pretrained network, i.e. $s_t(\mathbf{x}_t) \approx s_\theta(\mathbf{x}_t, t)$. Hence the per–coordinate score difference used in the Fisher–MI integrand is

$$\Delta s_{t,i} \;=\; s_{t,i}(\mathbf{x}_t \mid \mathbf{x}_0) \;-\; s_{\theta,i}(\mathbf{x}_t, t).$$

**Pixel-specific truncation via a learnable adapter.** Instead of learning a full perturbation generator, we introduce a small U-Net $\tau_\phi$ that maps an input image batch $\mathbf{x}_0 \in \mathbb{R}^{B \times C \times H \times W}$ to a tensor

of per-pixel truncation times $\mathbf{t} = \tau_\phi(\mathbf{x}_0) \in [0,1]^{B \times C \times H \times W}$ (obtained via a sigmoid), one time upper bound $t_i$ per coordinate. Given $\mathbf{t}$, we *construct* a perturbed sample $\mathbf{x}'$ by elementwise forward noising with pixelwise times:

$$\mathbf{x}' = \alpha(\mathbf{t}) \odot \mathbf{x}_0 + \sigma(\mathbf{t}) \odot \boldsymbol{\epsilon}, \qquad \boldsymbol{\epsilon} \sim \mathcal{N}(\mathbf{0}, \mathbf{I}),$$

where $\alpha(\mathbf{t})$ and $\sigma(\mathbf{t})$ are broadcast from the scalar schedules to the pixelwise times and $\odot$ denotes Hadamard product. The classifier $f$ is kept fixed; we use $f(\mathbf{x}')$ to enforce prediction preservation.

**Discretizing the Fisher–MI lower bound.** Assigning pixel-specific upper limits $t_i \leq 1$ yields a lower bound on the total MI. To estimate the per-coordinate integral we apply a change of variables $t = u\, t_i$ and use a $K$-point quadrature on $u \in [0,1]$. Let $\{u_k, \bar{w}_k\}_{k=1}^{K}$ be fixed quadrature nodes and *normalized* weights on $[0,1]$ (e.g., trapezoidal or Gauss–Legendre), with $\sum_k \bar{w}_k = 1$. For each pixel $i$, define $t_{i,k} := u_k\, t_i$. Then

$$C_i(t_i) = \frac{1}{2} \int_0^{t_i} \sigma(t)^2\, \mathbb{E}\big[(\Delta s_{t,i})^2\big]\, dt \approx \frac{t_i}{2} \sum_{k=1}^{K} \bar{w}_k\, \sigma(t_{i,k})^2\, \mathbb{E}\big[(\Delta s_{t_{i,k},i})^2\big].$$

Aggregating over coordinates gives

$$L_{\mathrm{MI}} = \sum_{i=1}^{d} C_i(t_i) \leq \frac{1}{2} \int_0^1 \sigma(t)^2\, \mathbb{E}\big[\|s_t(\mathbf{x}_t \mid \mathbf{x}_0) - s_\theta(\mathbf{x}_t, t)\|^2\big]\, dt = I(X_1; X_0).$$

In practice, the expectation is estimated by a minibatch average. Here $\bar{w}_k$ are the *quadrature weights on $[0,1]$* (e.g., for the trapezoidal rule $\bar{w}_1 = \bar{w}_K = \frac{1}{2}$ and $\bar{w}_k = 1$ otherwise, normalized to sum to 1; Gauss–Legendre rules use precomputed nodes/weights).

**Training objective and loop.** We optimize $\phi$ to minimize the loss function in Eq. equation 16. Each iteration samples $\boldsymbol{\epsilon}$, constructs $\mathbf{x}'$ using the current $\mathbf{t} = \tau_\phi(\mathbf{x}_0)$, draws quadrature knots $\{t_{i,k}\}$, evaluates $\Delta s_{t_{i,k},i}$ using the closed-form conditional score and the frozen $s_\theta$, accumulates $L_{\mathrm{MI}}$, adds $L_{\mathrm{CE}}$, and updates $\phi$. The pretrained $s_\theta$ and the classifier $f$ remain fixed throughout.

## 5 EXPERIMENTS

In this section, we systematically evaluate our method by comparing it against several benchmark attribution approaches. For baseline methods, we selected IBA Schulz et al. (2020), InputIBA Zhang et al. (2021), Integrated Gradients Sundararajan et al. (2017), Guided-BP Springenberg et al. (2014), DeepLIFT Shrikumar et al. (2017), and HSIC-Attribution Novello et al. (2022). In early exploratory tests, both RISE and DeepSHAP showed substantially weaker performance—RISE obtains very poor Insertion/Deletion AUCs in Xplique's official tutorial Fel et al. (2022), and DeepSHAP ranks near the bottom in InputIBA's evaluation (Fig. 5a)—so we did not include them as primary baselines. And the three methods (Guided-BP, Integrated Gradients, DeepLIFT) were implemented with Captum Kokhlikyan et al. (2020) in PyTorch Paszke et al. (2019), relying on the default parameter settings. For IBA, InputIBA, and HSIC-Attribution, we ran the official code releases without altering their recommended configurations. Our experiments are conducted on Ima-

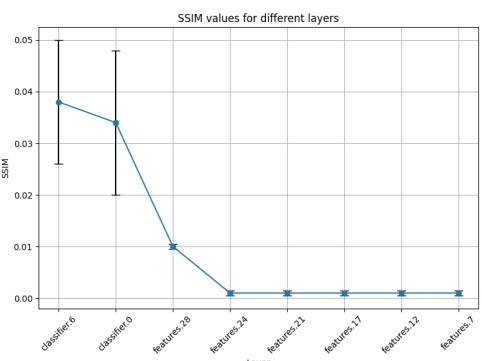

Figure 1: Parameter Randomization Sanity Check results. Our results show that SSIM values decrease sharply as randomization moves to earlier layers, demonstrating that our method effectively responds to parameter modifications and robustly identifies essential features.

geNet, as it is a standard dataset frequently employed in prior work on attribution (e.g., InputIBA Zhang et al. (2021). For the classifier, we use VGG16, which is widely used in attribution studies Schulz et al. (2020); Zhang et al. (2021) and provides a reliable benchmark. We additionally include experiments on ResNet-50 He et al. (2016) and EfficientNet Tan et al. (2019) and the results are provided in the Appendix.

## 5.1 PARAMETER RANDOMIZATION SANITY CHECK

The Parameter Randomization Sanity Check Adebayo et al. (2018) aims to assess whether attribution methods reliably explain model behavior by analyzing their sensitivity to parameter changes. This evaluation is performed using the Structural Similarity Index Metric (SSIM Wang et al. (2004)). A lower SSIM value between the attribution map of the original model and that of a randomized model indicates that the method is sensitive to parameter changes, effectively capturing key features influencing the model's decisions. Our experimental results demonstrate consistently low SSIM values across all layers where parameter randomization begins, highlighting the robustness of our approach in identifying essential features. We show the results of SSIM in Figure 1.

## 5.2 QUALITATIVE COMPARISON

The qualitative results in Figure 2 highlight clear differences in attribution quality across methods. IBA and HSIC-Attribution produce smooth but overly blurred maps, lacking detail and obscuring object boundaries. DeepSHAP and Integrated Gradients offer more fine-grained attributions but tend to be noisy and spatially incoherent, making it hard to identify meaningful patterns.

InputIBA improves IBA by sharpening focus areas, yet still suffers from edge leakage and background noise. Guided-BP generates visually sharp and textured maps, but often overemphasizes high-frequency details, misattributing importance to irrelevant regions.

In contrast, our method yields clean, well-structured attribution maps that preserve object contours while minimizing noise. By balancing detail and spatial coherence, it better isolates the most relevant features, offering clearer and more reliable visual explanations.

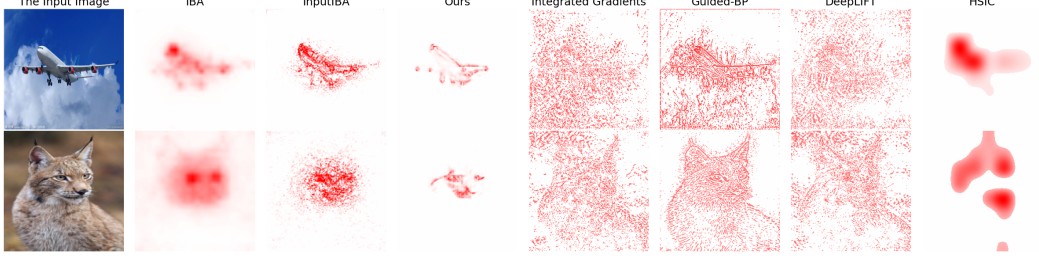

Figure 2: Qualitative comparison of attribution maps generated by different methods. IBA and HSIC-Attribution produce smooth but overly blurred attributions, failing to capture fine-grained details. Integrated Gradients and DeepLIFT generate highly detailed attributions but appear chaotic, lacking clear spatial coherence. Guided-BP enhances edge sharpness but often misallocates importance to irrelevant background regions. InputIBA improves upon IBA but still struggles with clearly delineating object boundaries. In contrast, our method produces well-defined and focused attributions, accurately highlighting the most relevant features while minimizing noise, demonstrating superior interpretability and robustness.

## 5.3 INSERTION AND DELETION AUCS

The Deletion and Insertion methods Zhang et al. (2021) evaluate attribution performance by progressively removing or adding pixels based on importance scores, with predictions monitored at each step to compute the Area Under the Curve (AUC). A larger Difference between insertion and deletion AUCs (DifAUCs) reflects better attribution quality. As shown in Table 1, our method achieves the highest DifAUCs scores, outperforming baseline methods.

| Method | DifAUCs |
|---|---|
| IBA | $0.771 \pm 0.006$ |
| InputIBA | $0.833 \pm 0.002$ |
| Integrated Gradients | $0.153 \pm 0.004$ |
| Guided-BP | $0.151 \pm 0.007$ |
| DeepLIFT | $0.157 \pm 0.008$ |
| Ours | $\mathbf{0.836} \pm 0.003$ |
| HSIC-Attribution | $0.133 \pm 0.002$ |

Table 1: Insertion and deletion experiments. Our method achieves the best performance, outperforming all baselines.

## 5.4 QUANTITATIVE VISUAL EVALUATION VIA EFFECTIVE HEAT RATIOS (EHR)

To further validate the effectiveness of our attribution method, we conduct a quantitative visual evaluation using Effective Heat Ratios (EHR Zhang et al. (2021)), but replacing the bounding box with segmentation annotations. This metric assesses the concentration of attribution scores within meaningful regions, providing a structured way to compare feature importance across different methods. We perform this evaluation on the FSS-1000 dataset Li et al. (2020), leveraging its high-quality segmentation annotations to establish ground truth regions of interest. The EHR metric quantifies the proportion of attribution scores assigned to these annotated regions, where a higher EHR indicates that an attribution method successfully localizes important features while minimizing noise in less relevant areas.

| Method | EHR |
|---|---|
| IBA | $0.355 \pm 0.004$ |
| InputIBA | $0.466 \pm 0.005$ |
| Integrated Gradients | $0.160 \pm 0.003$ |
| Guided-BP | $0.273 \pm 0.007$ |
| DeepLIFT | $0.155 \pm 0.008$ |
| Ours | $\textbf{0.587} \pm 0.006$ |
| HSIC-Attribution | $0.121 \pm 0.003$ |

Table 2: EHR experiment results. Our method achieves the highest Effective Heat Ratio (EHR), significantly outperforming all baselines. InputIBA ranks second, while other methods exhibit substantially lower EHR values.

By comparing our approach with baseline attribution methods, the results in Table 2 demonstrate that our method achieves the highest Effective Heat Ratio (EHR), indicating its superior ability to focus attributions on semantically meaningful regions.

## 5.5 BOUNDING BOXES EVALUATION

We adopt the bounding box evaluation introduced in Schulz et al. (2020), which measures the proportion of attribution mass that falls within ground-truth bounding boxes on ImageNet. Specifically, we compute the Box-Ratio, which measures the proportion of attribution mass that falls within human-annotated bounding boxes in the ImageNet dataset. A higher Box-Ratio indicates better localization performance, reflecting how well the explanation aligns with regions relevant to the target object. We make modifications to the algorithm; please refer to the Appendix H for details.

As shown in Table 3, our method achieves the highest Box-Ratio, outperforming all other baselines. This result confirms that our approach not only produces visually coherent attribution maps but also accurately aligns important regions with the object of interest, demonstrating strong localization capabilities.

## 6 CONCLUSION

We present a novel framework for feature attribution by modeling the perturbation space as a continuous-time stochastic differential equation (SDE). This general and unbounded formulation overcomes the limitations of fixed-scale perturbation methods and enables a principled exploration of input variations. By establishing a theoretical connection between KL divergence, Fisher divergence, and mutual information under time-dependent SDEs, we simplify the attribution objective into a tractable form. Extensive qualitative and quantitative experiments demonstrate that our approach produces more focused, coherent, and accurate attribution maps compared to existing baselines. By balancing fine-grained detail with

| Method | Box-Ratio |
|---|---|
| IBA | $0.997 \pm 0.001$ |
| InputIBA | $0.997 \pm 0.001$ |
| **Ours** | $\textbf{0.998} \pm \textbf{0.001}$ |
| Integrated Gradients | $0.691 \pm 0.006$ |
| Guided-BP | $0.698 \pm 0.002$ |
| DeepLIFT | $0.695 \pm 0.005$ |
| HSIC-Attribution | $0.903 \pm 0.002$ |

Table 3: Box-Ratio evaluation on ImageNet. A higher score indicates better localization of attributions within annotated bounding boxes. Our method achieves the highest Box-Ratio.

spatial consistency, our method offers a reliable and theoretically grounded tool for interpreting deep neural networks.

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

## A ALTERNATIVE DERIVATION (FOR COMPLETENESS)

We follow the proof structure of Lyu (2012) to enhance readability and make it easier for readers to identify which parts of our derivation are directly inherited from prior work and which parts are different.

*Proof.* To simplify notation and improve readability, explicit references to variables (e.g., $\mathbf{x}$, $\mathbf{y}$) and integration measures (e.g., $d\mathbf{y}$) are omitted throughout the proof whenever their omission does not cause ambiguity. The proof proceeds as follows.

By applying the Lemma 1 in Lyu (2012) , the Fisher divergence can be expressed as:

$$D_F(p_t\|q_t) = \int p_t\big(\|\nabla \log p_t\|^2 + \|\nabla \log q_t\|^2 + 2\Delta \log q_t\big). \tag{17}$$

Further simplifications lead to:

$$D_F(p_t\|q_t) = \int p_t\big(\|\nabla \log p_t\|^2 + \frac{\Delta q_t}{q_t} + \Delta \log q_t\big). \tag{18}$$

**Expanding the Time Derivative of $D_{\mathrm{KL}}(p_t\|q_t)$:**

We then begin by expanding the time derivative of the KL divergence:

$$\frac{d}{dt}D_{\mathrm{KL}}(p_t\|q_t) = \int \frac{\partial p_t}{\partial t} \log \frac{p_t}{q_t} + \int \frac{\partial p_t}{\partial t} - \int \frac{\partial p_t}{\partial t} \log q_t. \tag{19}$$

**Elimination of the Second Term:**

The second term vanishes because the integral of $p_t$ over the entire space is constant (due to normalization):

$$\int \frac{\partial p_t}{\partial t} = \frac{\partial}{\partial t} \int p_t = \frac{\partial}{\partial t}(1) = 0. \tag{20}$$

Thus, we have:

$$\frac{d}{dt}D_{\mathrm{KL}}(p_t\|q_t) = \int \frac{\partial p_t}{\partial t} \log \frac{p_t}{q_t} - \int \frac{\partial p_t}{\partial t} \log q_t. \tag{21}$$

Using Fokker–Planck equation:

$$\frac{\partial p_t}{\partial t} = -\nabla \cdot (\mu(t)p_t) + \frac{1}{2}\sigma(t)^2 \Delta p_t, \tag{22}$$

we decompose $\frac{d}{dt}D_{\mathrm{KL}}(p_t\|q_t)$ into two main terms:

$$\frac{d}{dt}D_{\mathrm{KL}}(p_t\|q_t) = I_1 + \frac{1}{2}\sigma(t)^2 I_2, \tag{23}$$

where the drift term $I_1$ and diffusion term $I_2$ are defined as:

$$I_1 = -\mu(t) \int \nabla p_t \log p_t + \mu(t) \int \nabla p_t \log q_t - \int p_t \frac{\partial}{\partial t} \log q_t, \tag{24}$$

$$I_2 = \int \Delta p_t \log p_t - \int \Delta p_t \log q_t - \int \Delta p_t \log q_t. \tag{25}$$

**Simplifying $I_1$:**

By integration by parts, $\int \nabla p_t \log p_t = 0$, so:

$$I_1 = -\mu(t) \int p_t \frac{\nabla q_t}{q_t} - \int p_t \frac{d}{dt} \log q_t. \tag{26}$$

Using the chain rule and Fokker–Planck equation:

$$\frac{d}{dt} \log q_t = -\mu(t)\frac{\nabla q_t}{q_t}, \tag{27}$$

we find:

$$I_1 = -\mu(t) \int p_t \frac{\nabla q_t}{q_t} + \mu(t) \int p_t \frac{\nabla q_t}{q_t} = 0. \tag{28}$$

**Simplifying $I_2$:** Using integration by parts:

$$\int \Delta p_t \log p_t = -\int p_t \|\nabla \log p_t\|^2, \tag{29}$$

$$\int \Delta p_t \log q_t = \int p_t \Delta \log q_t. \tag{30}$$

Thus:

$$I_2 = -\int p_t \|\nabla \log p_t\|^2 - \int p_t \Delta \log q_t - \int p_t \frac{\Delta q_t}{q_t}. \tag{31}$$

Using the definition of $D_F(p_t \| q_t)$:

$$I_2 = -D_F(p_t \| q_t). \tag{32}$$

**Combining Results**:

Since $I_1 = 0$, we have:

$$\frac{d}{dt} D_{\text{KL}}(p_t \| q_t) = -\frac{\sigma(t)^2}{2} D_F(p_t \| q_t), \tag{33}$$

which completes the proof. $\square$

## B  QUALITATIVE COMPARISON OF DIFFERENT CLASSIFIERS

To evaluate the performance of our attribution technique, we applied it to several convolutional neural network models—namely VGG16, ResNet50, and EfficientNet. The resulting attribution visualizations, presented below, illustrate how each architecture uniquely interprets and emphasizes different input features. The variation observed in these visual explanations highlights the distinct computational strategies inherent to each network. VGG16 and ResNet50 tend to distribute attention broadly across input regions, reflecting a preference for exhaustive feature representation. However, compared to VGG16, ResNet50 exhibits a noticeably higher reliance on a larger number of input pixels. In contrast, EfficientNet demonstrates a more selective focus, concentrating on specific, high-importance areas. This contrast provides valuable insights into model decision-making and guides the choice of architecture according to task-specific priorities such as prediction accuracy, explainability, and resource constraints.

## C  SUPPORT FOR OTHER DIFFUSION MODELS

Variance Preserving (VP) Process (Song et al., 2020) is defined by:

$$dX_t = -\frac{1}{2}\beta(t)X_t\,dt + \sqrt{\beta(t)}\,dW_t. \tag{34}$$

We typically define $\beta(t)$ as:

$$\beta(t) = \beta_{\min} + t(\beta_{\max} - \beta_{\min}), \tag{35}$$

with $\beta_{\min} = 0.1$ and $\beta_{\max} = 20$ (Song et al., 2020). Then $X_t$ can be expressed as:

$$X_t = X_0 e^{-\frac{1}{2}\int_0^t \beta(s)ds} + \sqrt{1 - e^{-\int_0^t \beta(s)ds}}\epsilon, \tag{36}$$

where $\epsilon \sim \mathcal{N}(0, \mathbf{I})$. We can explicitly compute the integral $\int_0^t \beta(s)ds$ due to the linear form of $\beta(s)$:

$$\int_0^t \beta(s)ds = \beta_{\min}t + \frac{1}{2}(\beta_{\max} - \beta_{\min})t^2. \tag{37}$$

For the Variance Preserving (VP) Process, we leverage the relationships derived in Denoising Diffusion Probabilistic Models (DDPM), which is the discrete-time formulation of the VP process. In DDPM, the forward diffusion process gradually adds Gaussian noise to the data over $T$ timesteps,

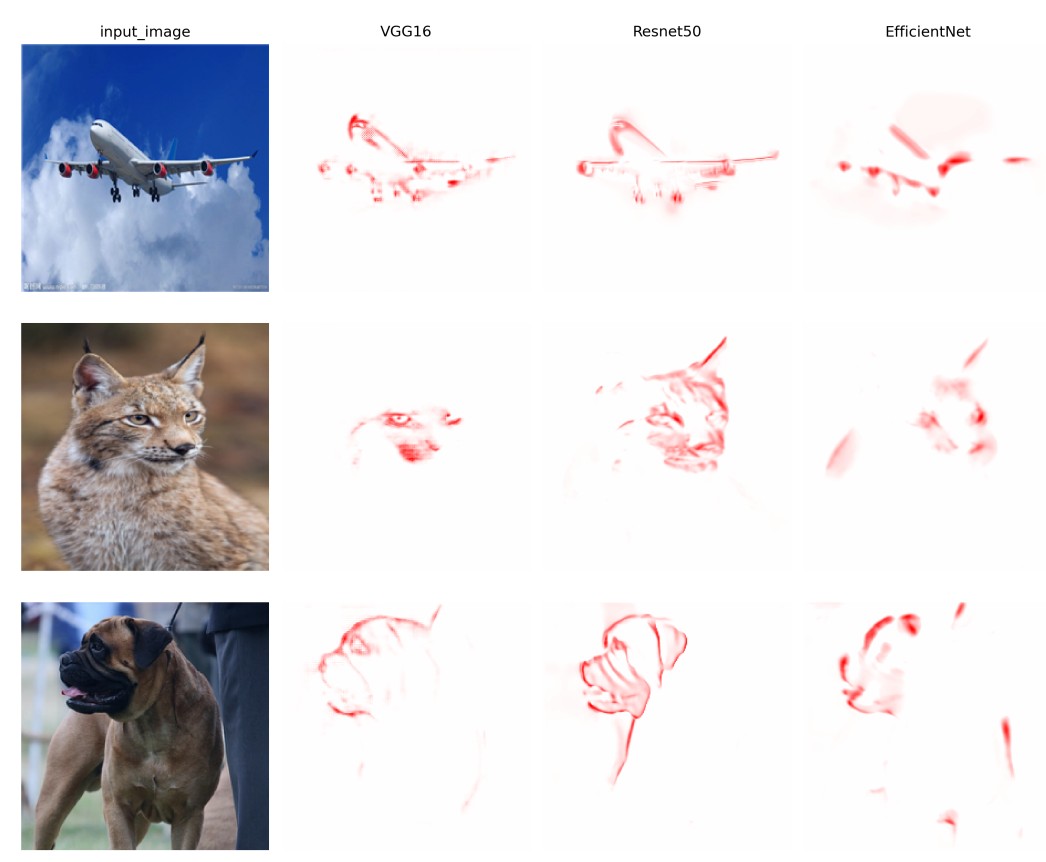

Figure 3: Attribution maps generated by VGG16, ResNet50, and EfficientNet, revealing model-specific patterns of feature prioritization.

resulting in noisy samples $X_t$. These relationships allow us to express $X_t$ and $X_0$ in terms of each other and the noise components.

The forward diffusion process in DDPM is defined as:

$$X_t = \sqrt{\bar{\alpha}_t}\, X_0 + \sqrt{1 - \bar{\alpha}_t}\, \epsilon, \quad \epsilon \sim \mathcal{N}(0, \mathbf{I}), \tag{38}$$

where $\bar{\alpha}_t = \prod_{s=1}^{t} \alpha_s$, and the relationship between the noise scheduling parameters $\alpha_t$ and $\beta_t$ is given by:

$$\alpha_t = 1 - \beta_t. \tag{39}$$

The conditional distribution $p_t(X_t \mid X_0)$ is Gaussian with mean $\sqrt{\bar{\alpha}_t}\, X_0$ and variance $(1 - \bar{\alpha}_t)\, \mathbf{I}$. Therefore, the conditional score function can be computed analytically:

$$s_t(X_t \mid X_0) = \nabla_{X_t} \log p_t(X_t \mid X_0) = -\frac{X_t - \sqrt{\bar{\alpha}_t}\, X_0}{1 - \bar{\alpha}_t}. \tag{40}$$

To approximate the marginal score function $s_t(X_t) = \nabla_{X_t} \log p_t(X_t)$, we train a neural network $\epsilon_\theta(X_t, t)$ to predict the noise $\epsilon$ given $X_t$, as proposed in DDPM. The marginal score function can then be approximated as:

$$s_t(X_t) \approx -\frac{1}{\sqrt{1 - \bar{\alpha}_t}}\, \epsilon_\theta(X_t, t). \tag{41}$$

By substituting $X_t$ from Eq. equation 38 into Eq. equation 40, we simplify the conditional score function:

$$s_t(X_t \mid X_0) = -\frac{\sqrt{1 - \bar{\alpha}_t}\, \epsilon}{1 - \bar{\alpha}_t} = -\frac{\epsilon}{\sqrt{1 - \bar{\alpha}_t}}. \tag{42}$$

With both $s_t(X_t)$ and $s_t(X_t \mid X_0)$ expressed in terms of $\epsilon$ and $\epsilon_\theta(X_t, t)$, then we have:

$$\|s_t(X_t) - s_t(X_t \mid X_0)\|^2 \approx \frac{1}{1 - \bar{\alpha}_t} \|\epsilon_\theta(X_t, t) - \epsilon\|^2. \tag{43}$$

## D  Notation Table

For clarity and ease of reference, we provide a comprehensive table of all mathematical notations used throughout this paper.

| Notation | Definition |
|---|---|
| **Model & Data** | |
| $f : \mathbb{R}^n \to \mathbb{R}$ | Machine learning model mapping input to output |
| $\mathbf{x} = [x_1, x_2, \ldots, x_n]^\top$ | Input feature vector |
| $\mathbf{x}'$ | Perturbed input |
| $X$ | Original data |
| $X'$ | Perturbed data |
| $Y$ | Label distribution |
| $n$ or $d$ | Input dimension |
| **Perturbation Parameters** | |
| $\mu_i$ | Multiplicative scaling factor for feature $i$ |
| $\delta_i$ | Additive noise term for feature $i$ |
| $D_i$ | Noise distribution for feature $i$ |
| $\xi$ | Threshold for output change constraint |
| **SDE Components** | |
| $X_t$ or $Y_t$ | Perturbed input at time $t$ |
| $\mu(t)$ | Time-dependent drift term |
| $\sigma(t)$ | Time-dependent diffusion coefficient |
| $dW_t$ | Standard Wiener process increment |
| $t$ | Diffusion time, $t \in [0, 1]$ |
| $t_i$ | Pixel-specific truncation time for coordinate $i$ |
| **Score Functions** | |
| $s_t(\mathbf{x}_t)$ | Marginal score function: $\nabla_{\mathbf{x}_t} \log p_t(\mathbf{x}_t)$ |
| $s_t(\mathbf{x}_t \mid \mathbf{x}_0)$ | Conditional score function: $\nabla_{\mathbf{x}_t} \log p_t(\mathbf{x}_t \mid \mathbf{x}_0)$ |
| $s_{t,i}(\cdot)$ | $i$-th coordinate of score function |
| $\Delta s_{t,i}$ | Score difference: $s_{t,i}(\mathbf{x}_t \mid \mathbf{x}_0) - s_{t,i}(\mathbf{x}_t)$ |
| $s_\theta(\mathbf{x}_t, t)$ | Pretrained neural network approximating marginal score |
| **Divergences & Information Measures** | |
| $D_{\mathrm{KL}}(p\|q)$ | Kullback-Leibler divergence between distributions $p$ and $q$ |
| $D_F(p\|q)$ | Fisher divergence: $\int p(x) \|\nabla \log p(x) - \nabla \log q(x)\|^2 \, dx$ |
| $I(X; X')$ | Mutual information between random variables $X$ and $X'$ |
| $i(\mathbf{x}; \mathbf{x}')$ | Pointwise mutual information between $\mathbf{x}$ and $\mathbf{x}'$ |
| **Probability Distributions** | |
| $p_t(\mathbf{x}_t)$ | Marginal probability density at time $t$ |
| $q_t(\mathbf{x}_t)$ | Reference probability density at time $t$ |
| $p_t(\mathbf{x}_t \mid \mathbf{x}_0)$ | Conditional probability density given initial state |
| $q_t(\mathbf{x}_t \mid \mathbf{x}_0)$ | Forward noising kernel: $\mathcal{N}(\alpha(t)\mathbf{x}_0, \sigma(t)^2\mathbf{I})$ |
| **Loss Functions** | |
| $\mathcal{L}$ or $L$ | Total loss function |
| $\mathcal{L}_{\mathrm{IB}}$ or $L_{\mathrm{IB}}$ | Information bottleneck loss |
| $\mathcal{L}_{\mathrm{MI}}$ or $L_{\mathrm{MI}}$ | Mutual information loss term |
| $\mathcal{L}_{\mathrm{CE}}$ or $L_{\mathrm{CE}}$ | Cross-entropy loss |
| $\beta$ | Trade-off hyperparameter balancing MI and CE losses |

Table 4: Notation table (Part 1 of 2)

## E  Extended Experimental Setup

To ensure reproducibility and transparency, we provide a comprehensive description of the experimental configurations employed in our study. This section details the hyperparameter settings for baseline methods, justifies our parameter choices, and describes the pretrained models utilized.

| Notation | Definition |
|---|---|
| **Attribution Components** | |
| $C_i(t_i)$ | Contribution of pixel $i$ up to time $t_i$: $\frac{1}{2}\int_0^{t_i}\sigma(t)^2\mathbb{E}[(\Delta s_{t,i})^2]dt$ |
| **Network Parameters** | |
| $\tau_\theta$ or $\tau_\phi$ | U-Net predicting pixel-specific truncation times |
| $\theta$ or $\phi$ | Learnable parameters of the adapter network |
| $\epsilon_\theta(\mathbf{x}_t, t)$ | Neural network predicting noise in DDPM |
| **Quadrature Components** | |
| $K$ | Number of quadrature points |
| $u_k$ | $k$-th quadrature node on $[0, 1]$ |
| $\bar{w}_k$ | Normalized quadrature weight for node $k$ (with $\sum_k \bar{w}_k = 1$) |
| $t_{i,k}$ | Scaled time for pixel $i$ at quadrature point $k$: $u_k \cdot t_i$ |
| **Diffusion Schedule (VP Process)** | |
| $\alpha(t)$ | Signal coefficient at continuous time $t$ |
| $\alpha_t$ | Discrete-time signal coefficient: $1 - \beta_t$ |
| $\bar{\alpha}_t$ | Cumulative product: $\prod_{s=1}^t \alpha_s$ |
| $\beta(t)$ | Continuous-time noise schedule function |
| $\beta_t$ | Discrete-time noise schedule at step $t$ |
| $\beta_{\min}, \beta_{\max}$ | Minimum and maximum noise levels in VP process |
| **Diffusion Schedule (VE Process)** | |
| $\sigma_{\min}, \sigma_{\max}$ | Minimum and maximum diffusion coefficients in VE process |
| **Noise Terms** | |
| $\epsilon$ or $\boldsymbol{\epsilon}$ | Gaussian noise sampled from $\mathcal{N}(\mathbf{0}, \mathbf{I})$ |
| $\mathbf{z}$ | Noisy sample: $\mathbf{x} + \sigma(\mathbf{t}) \odot \boldsymbol{\epsilon}$ |
| **Mathematical Operations** | |
| $\odot$ | Hadamard (element-wise) product |
| $\nabla$ | Gradient operator |
| $\Delta$ | Laplacian operator: $\sum_{i=1}^d \frac{\partial^2}{\partial x_i^2}$ |
| $\|\cdot\|$ | Euclidean norm (unless otherwise specified) |
| $\mathbb{E}[\cdot]$ | Expectation operator |
| $\nabla\cdot$ | Divergence operator |
| **Evaluation Metrics** | |
| SSIM | Structural Similarity Index Metric |
| DifAUCs | Difference between Insertion and Deletion AUCs |
| EHR | Effective Heat Ratio |
| Box-Ratio | Proportion of attribution mass within ground-truth bounding boxes |
| **Dimensions and Indices** | |
| $B$ | Batch size |
| $C$ | Number of channels |
| $H$ | Image height |
| $W$ | Image width |
| $i$ | Index for input feature/pixel coordinate |
| $k$ | Index for quadrature points |

Table 5: Notation table (Part 2 of 2)

We carefully configured each baseline attribution method following either their official implementations or established best practices in the literature. For Integrated Gradients Sundararajan et al. (2017), we configure the approximation with 200 integration steps (`n_steps=200`), which provides a fine-grained approximation of the path integral from a baseline to the input. For Guided-GradCAM Selvaraju et al. (2017b), we specify the target convolutional layer as `model.features[30]` for VGG16, which corresponds to a high-level convolutional layer capturing semantic features relevant to the final classification decision. DeepLIFT Shrikumar et al. (2017) and Guided-BP Springenberg et al. (2014) are configured using their default settings in Captum Kokhlikyan et al. (2020). For IBA Schulz et al. (2020), we set the bottleneck capacity parameter $\beta_{\text{feat}} = 10$ and use the Adam optimizer with a learning rate of $5 \times 10^{-5}$. For InputIBA Zhang et al. (2021), which introduces a dual bottleneck mechanism, we configure $\beta_{\text{feat}} = 10$ for the feature-level bottleneck and $\beta_{\text{input}} = 20$ for the input-level bottleneck, performing 60 optimization iterations with the same optimizer settings. HSIC-Attribution Novello et al. (2022) is implemented using the official code release with default hyperparameters.

Our method introduces two primary hyperparameters: the information bottleneck trade-off weight $\beta$ and the number of quadrature points $K$ for discretizing the Fisher–MI time integral. Following the empirical precedent established by IBA Schulz et al. (2020) and InputIBA Zhang et al. (2021), we set $\beta = 1$ uniformly across all experiments. To validate this choice, we conducted preliminary sensitivity analysis with alternative values ($\beta \in \{0.25, 0.5, 2, 4\}$). We observed that deviating from $\beta = 1$ resulted in substantial degradation of attribution quality: smaller values led to insufficient information compression with overly diffuse attribution maps, while larger values caused excessive compression with incomplete or fragmented attributions. Given the robustness of $\beta = 1$ across diverse inputs, we adopt this value as a fixed constant rather than a tunable hyperparameter.

The parameter $K$ determines the discretization granularity when approximating the continuous-time integral via numerical quadrature. We employ $K = 8$ quadrature points with normalized trapezoidal weights. This choice is motivated by empirical convergence analysis: increasing $K$ from 8 to 16 yields negligible improvement in attribution accuracy while doubling computational cost. Conversely, reducing $K$ below 8 introduces noticeable discretization artifacts. Thus, $K = 8$ provides an optimal balance between approximation fidelity and computational efficiency. Both $\beta$ and $K$ remain fixed across all experiments and generalize well to unseen samples without requiring per-instance tuning.

A key component of our method is the pretrained score-based diffusion model, which provides the marginal score function $s_\theta(\mathbf{x}_t, t)$ used to compute the score difference in the Fisher–MI integrand. We leverage publicly available pretrained models from the Hugging Face Diffusers library, which offers implementations of multiple diffusion variants pretrained on ImageNet. Specifically, we utilize models including both Variance Exploding (VE) and Variance Preserving (VP) formulations such as DDPM Ho et al. (2020) and Score-SDE Song et al. (2020). These models have been trained on ImageNet covering a wide range of visual domains and object categories, ensuring robust generalization to diverse inputs. The pretrained score networks remain frozen throughout our attribution procedure—only the lightweight adapter network $\tau_\phi$ is trained to predict pixel-specific truncation times. By building upon pretrained diffusion models, our approach inherits their strong inductive biases for natural image statistics while requiring minimal additional training.

All experiments are conducted on a single machine equipped with an NVIDIA RTX 4090 GPU, as stated in the main experiments section. The implementation uses PyTorch with CUDA acceleration for all methods.

# F  INSERTION AND DELETION AUCs: DETAILED METHODOLOGY

The Insertion and Deletion AUCs methodology provides a quantitative framework for evaluating attribution quality by measuring how model predictions respond to progressive pixel modifications based on attribution scores. Following the protocol established in InputIBA Zhang et al. (2021), our evaluation adopts the standard Deletion and Insertion procedures.

## F.1  DELETION PROCESS

The Deletion process evaluates attribution quality by progressively removing pixels in order of descending importance. At each step, the top-$k$ most important pixels (as indicated by highest attribution scores) are masked, and the model's prediction confidence for the original class is recorded. A good attribution method should identify truly important features, such that their removal causes rapid degradation in prediction confidence. The complete deletion curve traces how confidence decreases as more high-importance pixels are eliminated from the input.

## F.2  INSERTION PROCESS

Conversely, the Insertion process begins with an initially fully-masked image where all pixels are occluded. Pixels are then progressively revealed in order of descending importance based on attribution scores, with the top-$k$ most important pixels being added at each step. The model's prediction confidence is recorded after each insertion. An effective attribution method should identify the most critical features first, enabling the model to recover high confidence rapidly with minimal pixel ad-

ditions. The insertion curve captures how prediction confidence increases as important pixels are progressively restored.

## F.3 DifAUCs Computation

The quality of an attribution method is quantified using DifAUCs (Deletion AUCs minus Insertion AUC), computed as:

$$\text{DifAUCs} = \text{AUC}_{\text{Insertion}} - \text{AUC}_{\text{Deletion}} \tag{44}$$

where each AUC is obtained by integrating the prediction confidence curve over the progressive perturbation steps. A larger DifAUCs value indicates superior attribution performance: the method successfully identifies features whose removal substantially harms predictions (high Deletion AUC) while enabling rapid confidence recovery when added back (high Insertion AUC).

## F.4 Implementation Details

Our implementation follows standard practices established in the attribution literature:

- **Number of steps**: 100 progressive perturbation steps, following the standard protocol in InputIBA Zhang et al. (2021).
- **Pixels modified per step**: At each step, 1% of the total pixels are either masked (Deletion) or revealed (Insertion).
- **Masking strategy**: Masked pixels are set to 0, representing complete occlusion of the corresponding image regions.

This progressive perturbation protocol ensures fine-grained evaluation of attribution quality across the entire spectrum from fully intact to heavily corrupted inputs.

## F.5 Experimental Results

Figure 4 presents the Insertion and Deletion curves for all evaluated methods. The Insertion curves (Figure 4a) demonstrate that our method, along with IBA and InputIBA, achieves rapid confidence recovery, reaching high prediction confidence (¿0.9) within the first 20% of pixels added. In contrast, gradient-based methods (Integrated Gradients, Guided-BP, DeepLIFT, Guided-GradCAM) exhibit delayed recovery, maintaining near-zero confidence until over 80% of pixels are revealed, indicating poor identification of critical features.

The Deletion curves (Figure 4b) show complementary behavior: our method and the information bottleneck baselines (IBA, InputIBA) demonstrate sharp confidence degradation as important pixels are removed, while gradient-based methods exhibit slower decline, suggesting their attributions include many non-critical features. The quantitative DifAUCs scores reported in Table 1 confirm these observations, with our method achieving the highest DifAUCs ($0.836 \pm 0.003$), demonstrating superior attribution accuracy compared to all baselines.

# G EHR Computation Methodology

The Effective Heat Ratio (EHR) metric quantifies how well attribution scores concentrate within ground-truth bounding boxes across different threshold levels. Unlike previous metrics that only consider the rank of pixels, EHR accounts for the full distribution of attribution values both inside and outside the bounding box.

**Computation Procedure:** For each image with ground-truth bounding box annotations, we compute EHR through the following steps:

1. **Define quantile thresholds**: Generate a set of quantile values $Q = \{q_1, q_2, \ldots, q_K\}$ uniformly distributed in $[0, 1]$, where $K$ is the number of quantiles (In our experiments, we use $K = 11$ quantile thresholds uniformly spaced in the interval $[0.0, 1.0]$, specifically $Q = \{0.0, 0.1, 0.2, 0.3, 0.4, 0.5, 0.6, 0.7, 0.8, 0.9, 1.0\}$).

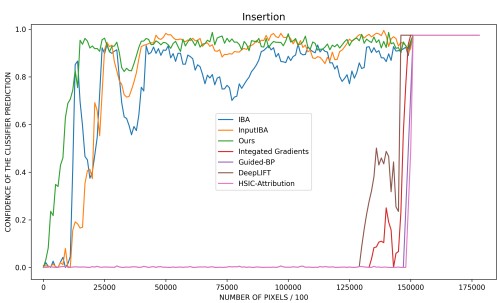 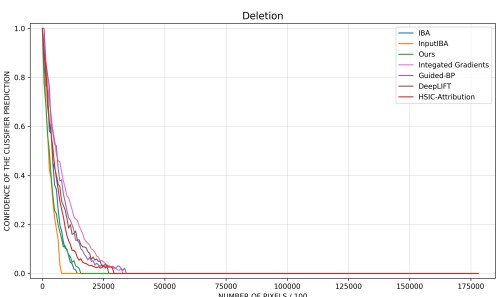

(a) Insertion curves showing prediction confidence recovery as important pixels are progressively revealed.

(b) Deletion curves showing prediction confidence degradation as important pixels are progressively removed.

Figure 4: Insertion and Deletion evaluation curves. Our method (green) demonstrates rapid confidence recovery in insertion and sharp degradation in deletion, indicating accurate identification of critical features.

2. **Compute threshold values**: For each quantile $q_k \in Q$, determine the corresponding attribution threshold $\theta_{q_k}$ as the $q_k$-th percentile of all attribution values in the image.

3. **Calculate effective heat ratio**: For each threshold $\theta_{q_k}$, compute the ratio:

$$\text{Ratio}(q_k) = \frac{\sum_{i \in \text{BBox}} \mathbb{1}[\text{attr}_i \geq \theta_{q_k}] \cdot \text{attr}_i}{\sum_{j=1}^{N} \mathbb{1}[\text{attr}_j \geq \theta_{q_k}] \cdot \text{attr}_j} \tag{45}$$

where $\text{attr}_i$ is the attribution score of pixel $i$, BBox denotes the set of pixels within the bounding box, $N$ is the total number of pixels, and $\mathbb{1}[\cdot]$ is the indicator function.

4. **Integrate over quantiles**: Compute the area under the curve of $\text{Ratio}(q)$ over all quantiles:

$$\text{EHR} = \frac{1}{K} \sum_{k=1}^{K} \text{Ratio}(q_k) \tag{46}$$

## H  BOUNDING BOX EVALUATION METHODOLOGY

The Bounding Box evaluation metric quantifies how well attribution methods localize important pixels within human-annotated object regions. Given an attribution map, we identify the top-$N$ most important pixels based on their attribution scores, where $N$ represents the minimal set of pixels required to maintain satisfactory model performance. We then compute the Box-Ratio as:

$$\text{Box-Ratio} = \frac{n}{N} \tag{47}$$

where $n$ is the number of top-$N$ attributed pixels that fall within the ground-truth bounding boxes, and $N$ is the total number of selected important pixels.

The computation proceeds as follows: First, we generate attribution maps for each image using the evaluated method. Second, we determine the value of $N$, which corresponds to the minimum amount of pixel information necessary to preserve prediction confidence. Third, we count how many of the top-$N$ pixels lie within the human-annotated bounding boxes that delineate the primary object in each image. Finally, we calculate the Box-Ratio by dividing $n$ by $N$. Note that bounding boxes may occasionally include background regions, particularly for non-rectangular objects, but they provide a reasonable proxy for semantically relevant areas.

## I  SEGMENTATION-BASED EVALUATION

While bounding box annotations provide a convenient proxy for object localization, they introduce inherent imprecision due to their rectangular shape. Bounding boxes inevitably include background

regions and non-object pixels, particularly for objects with irregular or non-rectangular geometries. This limitation can lead to inflated scores for attribution methods that scatter importance across both object and background areas within the box. To address this issue and provide a more rigorous evaluation of attribution quality, we introduce a segmentation-based assessment that leverages pixel-level annotations from the FSS-1000 dataset Li et al. (2020).

We define the Segmentation Ratio (SR) metric through the following procedure: (1) **Attribution Generation**: Generate attribution maps for each image and determine the minimum information threshold—the smallest number of pixel channel values needed to maintain satisfactory model performance. (2) **Overlap Computation**: Count how many of these most important pixels fall within the segmentation masks, denoting this overlap count as $n_{\text{seg}}$. (3) **Ratio Calculation**: Compute the SR as:

$$\text{SR} = \frac{n_{\text{seg}}}{N} \tag{48}$$

where $N$ is the total number of selected important pixels.

Table 6: Segmentation-based evaluation on FSS-1000. Information bottleneck methods achieve substantially higher scores than gradient-based approaches, with our method obtaining the best performance.

| Method | SR |
|---|---|
| IBA | $0.488 \pm 0.004$ |
| InputIBA | $0.468 \pm 0.003$ |
| **Ours** | $\mathbf{0.501 \pm 0.002}$ |
| Integrated Gradients | $0.079 \pm 0.006$ |
| Guided-BP | $0.078 \pm 0.009$ |
| DeepLIFT | $0.080 \pm 0.002$ |
| HSIC-Attribution | $0.377 \pm 0.005$ |

## J    THE USE OF LARGE LANGUAGE MODELS (LLMS)

In accordance with the ICLR policy on responsible use of large language models, we disclose that LLMs were employed solely to assist with the *polishing of language* in this manuscript. Specifically, LLMs were used to improve grammar, phrasing, and overall readability. No part of the research design, theoretical development, algorithmic contribution, or experimental analysis was generated by an LLM. All scientific ideas, derivations, and results presented in this paper are the sole work of the authors.

