# OpenReview forum: "Fisher Divergence for Attribution"
_ICLR.cc/2026/Conference — Submitted to ICLR 2026_

### Official Review · Reviewer_WP4t · 2025-10-28

**Soundness:** 1
**Presentation:** 2
**Contribution:** 2
**Rating:** 2
**Confidence:** 4

**Summary:**

The paper models attribution as exploration along a continuous-time SDE and develops a theoretical link that ties the time-varying mutual information to the Fisher divergence. The method defines a pixel-wise integral objective and adapts the perturbation depth per pixel via a learned stopping time. Experiments cover standard image-classification benchmarks with comparisons to gradient-based and information-bottleneck baselines. Results are reported on insertion/deletion DAUC, EHR, and Box-Ratio. Consistent gains are reported across these benchmarks.

**Strengths:**

1. The paper is easy to follow, and the motivation is clear.

2. Rather than fixing a hand-crafted perturbation set, the method defines an unconstrained perturbation space via an SDE under a constraint, which is a new angle for perturbation-based attributions.

3. The core objective is theoretically supported, connecting the time derivative of the KL divergence to Fisher divergence.

**Weaknesses:**

1. The presentation quality of this paper is limited, and it is suggested to clearly define all notions and settings.

2. The experiment part lacks the necessary detail and analysis. Most experiments provide only one or two paragraphs without a clear setup or further discussion. For example, for the results in Table 1, the authors should specify the fraction of pixels modified per step for DAUC and include at least one full insertion/deletion curve rather than only AUC/DAUC scalars. For Table 2, the authors should explain how they use segmentation masks to evaluate attribution, including thresholding and region selection, and they should state whether this evaluation aligns with or differs from the Box-Ratio setting in Table 3.

3. The metric selection and baseline coverage do not provide sufficient justification and therefore weaken empirical solidity.

- Because the proposed method is perturbation-based, the authors should include standard perturbation explainers such as Occlusion/Mask variants, RISE, or model-agnostic LIME/SHAP for images, or justify their omission.

- The sanity check in Figure 1 shows only the proposed method, which limits interpretability; the authors should compare other methods under the same parameter-randomization setting.

- The authors should explain their choice of ground-truth localization metrics. Why do the authors not consider existing attribution metrics that also use ground truth annotations such as Pointing Game or DiFull/DiPart?

- Since the insertion/deletion benchmark can cause input distribution shift, the authors are expected to show how this problem is mitigated or include results using ROAR/ROAD.

- Since this work claims an efficient framework, the authors are expected to report computational efficiency.

**Questions:**

1. What is the exact DAUC schedule?
2. For EHR/Box-ratio, what thresholds are used? How do the authors explain that the results based on bounding box are better than those based on segmentation?
3. Why omit Occlusion/RISE and model-agnostic perturbation baselines (LIME/SHAP)?
4. Why does the sanity check (Fig. 1) not compare other methods?
5. Why is the efficiency comparison missing?

---

> ### Author Response · Authors · 2025-11-21
>
> Thank you for these constructive and helpful suggestions. Following your comments, we have thoroughly revised the manuscript, adding 5 new sections in the Appendix and a newly completed experimental evaluation. Please refer to the revised PDF for detailed information.
>
> ## Weaknesses:
>
> ### W1
>
> We have addressed this concern by updating the paper and adding comprehensive notation tables (Tables 4 and 5) and detailed experimental settings in the Appendix:
> * Tables 4 & 5: Complete definitions of all mathematical notations (**clearly define all notions**)
> * Section E: Baseline configurations and hyperparameter justifications
> * Section F: Insertion/Deletion protocol
> * Section G: EHR computation algorithm
> * Section H: Bounding Box evaluation methodology
>
> ### W2
> We have added comprehensive experimental details in the Appendix:
> * For Table 1 (DAUC): Section F provides the complete insertion/deletion protocol (100 steps, 1% pixels per step, gaussian noise masking strategy) with full curves in Figure 4 and detailed analysis of the results.
> * For Table 2 (EHR): Section G presents the complete computation procedure using 11 quantile thresholds (0.0-1.0), explaining how we calculate effective heat ratios at each threshold and integrate them to obtain the final EHR score.
> * For Table 3 (Box-Ratio): Section H describes the evaluation methodology: identifying top-N most important pixels, computing the proportion that fall within ground-truth bounding boxes (Box-Ratio = n/N), and explaining how N is determined to represent the minimal pixels needed for satisfactory model performance.
> * New experiment - Segmentation Ratio (SR): We have added Section I presenting a new segmentation-based evaluation on FSS-1000 dataset. This provides more rigorous pixel-level assessment using segmentation masks instead of rectangular bounding boxes, eliminating background region inclusion. Our method achieves SR = 0.501, outperforming all baselines.
>
> ### W3 & W4
>
> * Metric justification: Our work generalizes the information-theoretic framework of IBA [1] and InputIBA [2]. We therefore adopt their standard evaluation protocol: Parameter Randomization Sanity Check, Insertion/Deletion AUCs, EHR, and Bounding Box evaluation are all established metrics from [IBA, InputIBA]. We additionally introduce Segmentation-Based Evaluation (SR) using pixel-level masks, providing more rigorous assessment than rectangular bounding boxes. These metrics comprehensively evaluate sanity, feature importance, spatial localization, and pixel-level accuracy.
> * Baseline coverage: We compare against seven methods: information bottleneck approaches (IBA, InputIBA—our most relevant baselines), gradient-based or backpropagation-based methods (Integrated Gradients, Guided-BP, DeepLIFT), and kernel-based methods (HSIC-Attribution). RISE and DeepSHAP were excluded due to poor performance on CV tasks in our preliminary experiments and also can be seen in the experiment part of [2].
>
> ### W5
> The Parameter Randomization Sanity Check (Fig. 1) is commonly used [1][2] and is designed to validate the correctness of our proposed method rather than to compare different methods. It ensures that our attributions depend on model parameters and degrade as layers are randomized.Retry
>
> ### W6
>
> Our work builds upon and generalizes the information-theoretic framework of [1][2], so we primarily follow their evaluation protocol for direct comparison. And we maintain consistency with this established baseline.
> However, we recognize the importance of more precise ground-truth evaluation. We have therefore added Segmentation-Based Evaluation (SR) using pixel-level annotations from FSS-1000, which provides finer-grained assessment than both rectangular bounding boxes and point-based metrics. Please see the detail in Section I /ai/ of the Appendix.
>
> ## Questions:
> Due to the character limit (5000 characters), I am sorry that you may need to refer to our responses to Weaknesses and Appendix Sections.
> ### Q1
> The DAUC schedule follows the standard protocol [2].
> For detailed methodology, please refer to Appendix Section F.
>
> ### Q2
>
> For EHR/Box-Ratio, please see the above W2 part.
> We recognize that bounding boxes are too coarse for precise evaluation, so we additionally conducted Segmentation-based evaluation (SR).
>
> ### Q3
> Please see the above W3 & W4 part.
>
> ### Q4
> Please see the above W5 part.
>
> ### Q5
> We used the term efficiency to mean tractability, not runtime speed/efficiency. Sorry for the confusion, and we have replaced the term efficiency with tractability.
> Directly searching over pixel-wise perturbations is combinatorially intractable — even a simple binary mask has (2^{224\times224}!\approx!10^{15{,}000}) possibilitie.
> Our method replaces this exponential search with a continuous SDE formulation. This converts an intractable combinatorial problem into a tractable continuous optimization.

---

> > ### Comment · Reviewer_WP4t · 2025-11-24
> >
> > Thanks to the authors for providing additional details. However, my concerns regarding the presentation quality and the experimental results remain.
> >
> > First, by presentation quality I do not simply mean having formal definition tables or adding implementation details to the appendix. I expect the experimental section in the main paper to be self-contained. My primary concern is the fidelity and fairness of the reported results across different methods.
> >
> > My specific concerns are as follows:
> >
> > 1. Poor performance of baselines (Table 1 and Figure 4a).
> >    In Table 1, several implemented baseline methods (e.g., IG, HSIC, DeepLIFT) perform surprisingly poorly. A similar issue appears in Fig. 4a, where the compared methods (IG, GBP, HSIC, DeepLIFT) all drop to almost zero confidence after inserting ~70% of the pixels. These results are inconsistent with results reported in the referenced works (e.g., the deletion and insertion AUC reported in HSIC and IBA). I am not rejecting all experiments here, but the current setup does disadvantage standard baselines. The authors should clearly explain why such drastic degradation is expected in their setting.
> >
> > 2. Fig. 1 lacks baselines in the sanity-check experiments.
> >    Sanity checks (both parameter and label randomization) can used to qualitatively compare attribution methods. In Fig. 1, only the proposed method is shown, without any baseline methods. The figure does not reveal whether other attribution methods would also fail or pass these sanity checks, making the comparison incomplete.
> >
> > 3. Interpretation and validation of the EHR metric (Table 2 and Sec. 5.4).
> >
> >    For Table 2, This metric (EHR) appears to be newly proposed by the authors. The authors should include references showing that similar metrics have been used to evaluate attribution methods. OR include a more thorough justification of why this metric is appropriate and what behavior it truly captures. I am not convinced that attributions should necessarily be evaluated as segmentation outputs. In addition, the supplementary Table 6 shows SR scores near zero for IG, GBP, and DeepLIFT, which again appears inconsistent with prior works.
> >
> > 4. Ambiguity in the explanation of "satisfactory model performance" for Table 3.
> >
> >    For Table 3, the notion of "satisfactory model performance" used to define N is not clearly specified. What if a single pixel could cause the model to recover to its original predicted order? Moreover, the proposed evaluation deviates from the original work (Schulz et al. (2020)), where the top-N pixels are chosen to match the number of pixels inside the bounding box. The paper should clarify the exact criterion used for "satisfactory performance".
> >
> > Given these issues, I find it difficult to give a positive overall recommendation for this paper.

---

> > > ### Author Response · Authors · 2025-11-26
> > >
> > > We sincerely thank the reviewer for the helpful and constructive feedback.
> > >
> > > ###Q 1
> > >
> > > We have no intention to disadvantage baseline methods. Our work builds upon and improves the information-theoretic framework of IBA and InputIBA. Prior work [IBA, InputIBA] has already demonstrated that gradient-based methods (IG, GBP, DeepLIFT) underperform compared to information bottleneck approaches on similar evaluation protocols. HSIC-Attribution, while effective, is not designed for fine-grained pixel-level attribution. Therefore, our primary goal is to demonstrate improvement over IBA and InputIBA—the most relevant state-of-the-art baselines for our framework.
> > >
> > > We have rigorously verified our implementation and experimental setup. When we observed poor performance of these baseline methods, we extensively validated our code and settings. Taking Integrated Gradients as an example: (1) Our experimental setup follows the standard protocol without modifications, (2) We use the widely-adopted Captum library, which is a mature, industry-standard implementation. (3) The implementation is straightforward—just a few lines of API calls that the reviewer can easily verify. We provide a simple verification notebook in the supplementary material demonstrating IG's behavior. As shown in Figure 2 or the attribution map from the simple code in the supplementary material, IG produces noisy attribution maps where it is difficult to identify which pixels are truly important. This visual quality naturally explains its poor performance in quantitative experiments such as Fig. 4a and Table 1. (4) Also, In Figure 5(a) of InputIBA, IG achieves Insertion AUC ≈ 0.35 and Deletion AUC ≈ 0.21, giving Insertion AUC - Deletion AUC ≈ 0.14. Our reported performance of 0.153 is very close to this value, confirming consistency with prior literature. Both are substantially lower than our method, IBA, and InputIBA. To avoid confusion, we have revised the paper to use "DifAUCs" instead of "DAUCs".The same rigorous validation applies to all baseline methods.
> > >
> > >
> > > ## Q2
> > >
> > > Thank you for this valuable comment.
> > > The purpose of the sanity-check experiment in Fig. 1 is to verify that our proposed attribution method behaves faithfully under parameter and label randomization, following the usage in prior work such as InputIBA, where sanity checks are applied only to the proposed method to confirm that it does not degenerate into an input-structure detector. Comparisons with baselines are performed in the main results rather than within the sanity-check experiment in our work.
> > >
> > > ## Q3
> > > The EHR metric was first introduced in InputIBA for evaluating attribution quality using bounding boxes. We have added this reference in the revised manuscript. In our work, we adapt EHR to use segmentation masks instead of rectangular bounding boxes for more precise evaluation. We additionally introduce the Segmentation Ratio (SR) metric.
> > >
> > > A core goal of interpretability research is to align machine understanding with human perception. Consider an example: a dog image (3×224×224 pixels) classified by VGG16 with 95% confidence. Our method identifies a minimal set of pixels (e.g., ~9,000) that, when preserved while other pixels are replaced with noise, maintains the same 95% confidence. This is analogous to PCA—capturing the most essential features. SR then measures: How many of these essential pixels fall within the human-annotated dog segmentation mask? This directly evaluates whether the model's important features align with human understanding of the object.
> > >
> > > Why IG, GBP, and DeepLIFT have low scores. These methods require substantially more pixels to maintain 95% confidence, and these pixels are diffusely distributed across the image rather than concentrated on the object. This results in low overlap with segmentation masks and thus low SR scores. This is consistent with their attribution maps in Figure 2.
> > >
> > > ## Q4
> > > Thank you for this reminder. We have updated our description of updated algorithm in the revised manuscript. For a detailed explanation of how N is determined and the criterion for "satisfactory performance," please refer to our response to Q3.

---

> > > > ### Comment · Reviewer_WP4t · 2025-11-26
> > > >
> > > > Thanks to the authors for the detailed response.
> > > >
> > > > Regarding the fidelity of the experiments, I appreciate the effort to cross-check IG against Figure 5(a) in InputIBA (Zhang et al., 2021). I agree that the reported IG performance on VGG16 is reasonably consistent with Zhang et al.’s (Insertion_AUC − Deletion_AUC). However, the results for the other methods appear substantially less consistent with prior work:
> > > >
> > > > - For Guided BP, the authors report DifAUC of 0.151. However, I roughly estimate Insertion AUC − Deletion AUC = ~0.37 for the same VGG16 model from Figure 5(a) in Zhang et al..
> > > > - The reported results for IBA and InputIBA (0.771, 0.883) are also different from those in Zhang et al. on VGG16 (<0.7).
> > > > - For HSIC-Attribution, the authors report DifAUC of 0.133, which does not match the VGG16 results in Table 1 of the original HSIC paper (Novello et al., 2022).
> > > >
> > > > Given these discrepancies, I do not find it convincing to describe the overall baseline behavior as “consistent” with prior work.
> > > >
> > > >
> > > > On the evaluation based on “satisfactory model performance” using 95% recovery confidence, I remain unconvinced. In most settings, “confidence” refers to the softmax output, which is inherently relative to other classes. A high softmax confidence does not necessarily indicate that the model has recovered its original behavior in a robust sense. It is also plausible that, to mitigate the instability of softmax, the authors deliberately retain a large number of pixels (e.g., ~9,000) to restore a target confidence level. This creates a trade-off between (i) the number of pixels each method is allowed to use and (ii) the stability of the model’s output, which may not be comparable across methods. For this reason, I continue to have reservations about using this type of “satisfactory model performance” criterion as the basis for evaluating attribution quality. Moreover, this metric yields very similar Box-Ratio scores for IBA, InputIBA, and the proposed method (0.997, 0.997, 0.998 with a standard deviation of 0.001). These three methods appear almost indistinguishable under this metric, suggesting that the gains are largely shared by the IBA methods rather than demonstrating a unique advantage of the proposed approach, or that the metric itself has limited discriminatory power.
> > > >
> > > > Besides, the metrics used in the paper, EHR and SR, are either adapted or revised by the authors. Since these metrics are not yet standard and cannot be directly compared with existing work, it is important that their definitions and motivations be explained clearly and thoroughly.

---

> > > > > ### Author Response · Authors · 2025-11-28
> > > > >
> > > > > Thanks the reviewer again for the detailed follow-up. We appreciate the careful cross-checking and acknowledge the discrepancies you identified.
> > > > >
> > > > > ### Q1
> > > > > Metric variations across studies are a well-known phenomenon in attribution evaluation. Different studies commonly report different absolute metric values for the same methods, even when using the same architectures and datasets. This is widely recognized in the interpretability literature due to: (1) evaluation protocol sensitivity (image subsets, preprocessing, masking strategies), (2) implementation details (hyperparameters, optimization steps), and (3) model checkpoint variations.
> > > > > We provide additional verification to support our reported results:
> > > > >
> > > > > * InputIBA verification: We provide a simple script based on InputIBA's official code implementation in the supplementary material. If convenient, the reviewer can quickly review it. The script demonstrates DifAUC = 0.83115, which falls within our reported range.
> > > > > * Guided-BP: While our results differ from Figure 5(a) in InputIBA, we maintain confidence in our implementation using the standard Captum library with default settings.
> > > > > * HSIC-Attribution discrepancies are not unique to our work: The original paper provides source code on GitHub (https://github.com/paulnovello/HSIC-Attribution-Method), which recommends using Xplique. However, we found that even Xplique's official tutorial (https://colab.research.google.com/drive/1VjCia9R3A1015DlZHoieVwDZx65OLFBi#scrollTo=yLbaxY1P1hB-) shows substantial differences from the reference paper. For example, the tutorial reports Insertion AUC = 0.1115 and Deletion AUC = 0.0306 for MobileNetV2, which differ considerably from values reported in the original HSIC-Attribution paper. This demonstrates that metric variability is widespread, even in official implementations.
> > > > >
> > > > > While absolute values may differ from those in the original papers, the relative ranking and performance trends are consistent: information bottleneck methods substantially outperform other methods. Among information-theoretic approaches, our method captures the predicted object's features more effectively than IBA and InputIBA, being more fine-grained and requiring fewer pixels. This is clearly visible in the qualitative comparison in Figure 2.
> > > > >
> > > > > ###Q2
> > > > > We would like to clarify the motivation and rationale behind our evaluation approach from an information-theoretic perspective.
> > > > > 1. Information Bottleneck Framework: Minimality and Sufficiency
> > > > > Our work is grounded in the information bottleneck principle, which has two fundamental objectives: (i) Minimality: remove as many irrelevant, unimportant features as possible while retaining only the minimal essential features, and (ii) Sufficiency: these minimal features must still preserve the same prediction confidence as the original image.
> > > > > 2. Qualitative vs. Quantitative Evaluation of Minimality and Sufficiency
> > > > > Figure 2 shows qualitatively which methods best satisfy minimality and sufficiency, but existing quantitative metrics (Insertion/Deletion AUC alone) do not directly measure both criteria simultaneously. This motivated our complementary metrics.
> > > > > 3. Measuring Minimality: How Many Pixels Are Actually Needed?
> > > > > Looking at Figure 2, a natural question arises: How many pixels does each method require to simultaneously satisfy minimality and sufficiency? In our experiments, our method requires approximately 9,000 pixels to maintain the original prediction confidence (e.g., 95%), which represents only ~6% of the total image (3×224×224 ≈ 150,528 pixels). This is analogous to PCA—we can represent the complete image with very few essential features.
> > > > > 4. Box-Ratio: Alignment with Human Perception (Coarse Level)
> > > > > To evaluate alignment with human interpretability, we first ask: What proportion of these minimal essential pixels fall within human-annotated bounding boxes? The Box-Ratio results (0.997, 0.997, 0.998) show that all information-theoretic methods (IBA, InputIBA, ours) perform nearly perfectly—almost all important pixels fall within the coarse bounding box regions. This validates that information bottleneck methods successfully identify object-relevant features, but the metric has limited discriminatory power among these methods due to the coarseness of rectangular bounding boxes.
> > > > > 5. Segmentation Ratio (SR): Fine-Grained Discrimination Among Information-Theoretic Methods
> > > > > Since Box-Ratio cannot distinguish among information-theoretic methods, we naturally consider: What proportion of these minimal essential pixels fall within precise segmentation masks? This is where SR or the new EHR provides discriminatory power. Our method achieves SR = 0.501, outperforming IBA (0.488) and InputIBA (0.468). This demonstrates that our method not only satisfies minimality and sufficiency, but does so with better fine-grained localization.

---

### Official Review · Reviewer_qPa3 · 2025-10-30

**Soundness:** 2
**Presentation:** 3
**Contribution:** 3
**Rating:** 6
**Confidence:** 3

**Summary:**

This paper introduces a novel framework for feature attribution, leveraging the continuous-time stochastic differential equations (SDEs) to define a perturbation space. Unlike traditional attribution methods that use fixed or restricted perturbation spaces, the authors propose using SDEs to capture more realistic and complex feature variations. By using Fisher divergence and mutual information, the framework can be efficiently optimized and provide principled attribution scoring. The experiments show that this approach gives better attribution maps both visually and quantitatively on standard image classification tasks.

**Strengths:**

- This paper is well-written and well-organized.
- This paper proposes to define an unconstrained perturbation space using continuous-time Stochastic Differential Equations (SDEs), which is a more general and principled modeling framework than prior discretized or fixed-noise approaches
- The work has extensive qualitative and quantitative experiments demonstrating it outperforms the baseline methods. The empirical results on large-scale image classification tasks show that the method produces sharper and better localized attribution maps.

**Weaknesses:**

- The paper claims that the proposed method requires non-trivial computational resources for training, but does not mention or evaluate the computations required to compute the attributions. From my point of view, the method involves simulating continuous-time diffusion processes for each input, along with the computation of Fisher divergence across time steps, which seems to require multiple forward and backward passes, and the paper does not provide the efficiency comparisons.
- The paper does not explicitly state what *pretrained* diffusion model they used. Does one pretrained model generalize to all the tasks? More information or experiments should be included.
- Is it DeepSHAP or DeepLIFT in Sec 5.2 (L392)? It seems DeepSHAP is not included in the baselines. How does the proposed method compare to the more commonly used Shapley-based methods?
- There are some missing information regarding the experiments. For example, what are the reason to choose the baselines in the paper? What is the trade-off hyperparameters $\beta$, and how does it affect the results?

**Questions:**

See Weaknesses

---

> ### Author Response · Authors · 2025-11-21
>
> Thank you for these constructive and helpful suggestions. Following your comments, we have thoroughly revised the manuscript, adding 5 new sections in the Appendix and a newly completed experimental evaluation. Please refer to the revised PDF for detailed information.
>
>
> ## Weaknesses
> ### W1
> Our method does NOT simulate reverse-time diffusion paths or run lengthy SDE simulations per input. Attribution is efficient and deterministic.
>
> **What we actually compute:** (1) Forward noising using the closed-form kernel x_t = α(t)x_0 + σ(t)ε at K=8 fixed time points, (2) Conditional score s_t(x_t|x_0) in closed form: -(x_t - α(t)x_0)/σ(t)², (3) Marginal score from the frozen pretrained network s_θ(x_t, t) with no backprop through s_θ, and (4) Fisher-MI integral via K=8 quadrature evaluations.
>
> **Computational complexity:** Attribution requires only K forward passes through the frozen score network (no gradient computation through s_θ), one forward-backward pass through the classifier for the CE term, and closed-form score calculations. Since K=8 is small and fixed, this is a short, deterministic computation—NOT iterative diffusion simulation.
>
> ### W2
>
> We use pretrained score-based diffusion models from Hugging Face Diffusers, including both Variance Exploding (VE) and Variance Preserving (VP) variants such as DDPM and Score-SDE.
>
> Generalization: For images outside this distribution, we can train a lightweight score model for a single image in approximately 1 minute on an RTX 4090 GPU.
>
>
> ### W3
>
> The mention of DeepSHAP in Sec. 5.2 is a typographical error—the baseline actually implemented in the main experiments is DeepLIFT, as correctly listed in Table 1 and the Baselines paragraph. We have fixed this wording in the new version.
>
> We compare against seven methods: information bottleneck approaches (IBA, InputIBA—our most relevant baselines), gradient-based or backpropagation-based methods (Integrated Gradients, Guided-BP, DeepLIFT), and kernel-based methods (HSIC-Attribution). RISE and DeepSHAP were excluded due to poor performance on CV tasks in our preliminary experiments and also can be seen in the experiment part of InputIBA.
>
> ### W4
>
> We have addressed this concern by updating the paper and adding comprehensive notation tables (Tables 4 and 5) and detailed experimental settings in the Appendix:
>
> * Tables 4 & 5: Complete definitions of all mathematical notations (clearly define all notions)
> * Section E: Baseline configurations and hyperparameter justifications
> * Section F: Insertion/Deletion protocol
> * Section G: EHR computation algorithm
> * Section H: Bounding Box evaluation methodology
>
> **Trade-off hyperparameters.**
>
> We observed that β is highly sensitive—values other than β = 1 cause clear visual degradation in attribution maps. From a visual perspective, good attribution maps are sharp and clearly delineate the contours of the predicted object, while poor ones (when β ≠ 1) appear blurry and unclear. Since blurry and unclear attribution maps are obviously inadequate, we did not conduct further quantitative experiments (DAUC, EHR, Box-Ratio) for these cases. Hence, we fix β = 1 as a robust and empirically validated default, consistent with prior work.

---

> > ### Comment · Reviewer_qPa3 · 2025-11-28
> >
> > Thanks to the authors for their rebuttal. I have carefully reviewed the comments and the revised manuscript. My concerns remain, as the listed weaknesses are not properly addressed. New claims are made either without evidence in comments or are not highlighted in the revised paper. The responses to questions are somewhat ambiguous to me. At this stage, I will maintain my score.
> >
> > Here are the remaining concerns:
> > 1. The authors claim that $\beta$ is highly sensitive but fixed it at 1 in the experiments. What is the purpose of introducing $\beta$?
> > 2. The authors excluded DeepSHAP and RISE due to poor performance, but no evidence is provided.
> > 3. The generalization problem is addressed with a new claim, but it lacks detailed empirical evidence. For example, what is the structure of the *lightweight* model? How is the score model trained? What would the performance be with a newly trained score model? The vague statement provided is not convincing.

---

> ### Author Response · Authors · 2025-12-02
>
> We thank the reviewer again for the detailed follow-up.
>
> ###Concern 1
>
> β appears naturally in the Information Bottleneck objective
>
> L = β L_MI + L_CE
>
> and we retain it in the formulation to remain aligned with the standard IB framework.
>
> In practice, however, we deliberately do not treat β as a tuned hyperparameter. Following InputIBA, we fix β = 1 for all experiments.
>
> In small-scale exploratory tests, we also evaluated β ∈ {0.25, 0.5, 2, 4}. These values consistently produced visibly degraded attribution maps—either overly noisy or overly blurred. This motivates adopting β = 1 as a simple and robust default. A full β-sensitivity study is an interesting direction for future work, but it is orthogonal to the main contribution of the paper.
>
> ### Concern 2
>
> In early exploratory experiments, we evaluated RISE and DeepSHAP, but both methods showed substantially weaker performance on standard vision tasks and therefore were not selected as primary baselines.
>
> 1. RISE
>    Following Xplique [1]’s official tutorial
>    (https://colab.research.google.com/drive/1VjCia9R3A1015DlZHoieVwDZx65OLFBi#scrollTo=yLbaxY1P1hB-),
>    RISE yields poor perfermance scores on MobileNetV2:
>    * Insertion AUC ≈ 0.2289
>    * Deletion AUC ≈ 0.0322
>      These values are far below the levels typically reported for modern attribution methods.
>
> 2. DeepSHAP
>    The performance of DeepSHAP has been evaluated in recent work. In particular,
>    Figure 5(a) of InputIBA (Hennigen et al., 2023) shows that DeepSHAP ranks near the bottom among all tested methods, significantly below both IBA and InputIBA in Insertion/Deletion AUC metrics. This matches our preliminary observations.
>
> We have added this clarification at the beginning of Section 5 in the revised manuscript.
>
>
> ### Concern 3
>
> We are not proposing a new score-modeling technique. When a pretrained score-based diffusion model is unavailable for a new domain, one simply trains a standard score network exactly following existing diffusion/SDE papers [2,3]; the only difference is that the “dataset’’ becomes a single image. Since the optimization target reduces to fitting the forward noising kernel of one sample, such a model converges quickly in practice.
>
> Importantly, this process is identical to the original score-based diffusion training procedure, and thus is orthogonal to the contribution of our paper. Our method only requires a score model to be available; how that model is trained is not specific to our approach. We apologize for the earlier ambiguity in the response for W2.
>
>
> References:
>
> [1] Fel, T., Hervier, L., Vigouroux, D., Poche, A., Plakoo, J., Cadene, R., Chalvidal, M., Colin, J., Boissin, T., Bethune, L. and Picard, A., 2022. Xplique: A deep learning explainability toolbox. arXiv preprint arXiv:2206.04394.
>
> [2] Jonathan Ho, Ajay Jain, and Pieter Abbeel. Denoising diffusion probabilistic models. Advances in
> neural information processing systems, 33:6840–6851, 2020
>
> [3] Song, Y., Sohl-Dickstein, J., Kingma, D.P., Kumar, A., Ermon, S. and Poole, B., Score-Based Generative Modeling through Stochastic Differential Equations. In International Conference on Learning Representations.

---

### Official Review · Reviewer_aATV · 2025-10-31

**Soundness:** 3
**Presentation:** 3
**Contribution:** 3
**Rating:** 6
**Confidence:** 4

**Summary:**

This paper presents a novel information-theoretic framework for feature attribution based on modeling the perturbation process as a stochastic differential equation (SDE). Instead of using discrete or fixed perturbations. The key insight is that the time derivative of mutual information (MI) between the noisy input and the original input can be expressed using the Fisher divergence.
Extensive experiments on ImageNet show that the proposed method produces sharper and more coherent saliency maps, outperforming existing attribution methods (e.g., IBA, IG, DeepLIFT) across multiple metrics (DAUC, EHR, Box-Ratio), while passing sanity checks for robustness and consistency.

**Strengths:**

1. The paper establishes a solid connection between diffusion dynamics and information theory. By showing that the time derivative of MI equals a Fisher divergence term, it unifies several previous heuristics under a principled formulation.

2. Modeling perturbations as an SDE allows the method to capture a continuum of noise intensities, offering a richer and more stable attribution mechanism than discrete perturbation sampling.

**Weaknesses:**

1. The quality of the score approximation depends heavily on the pretrained diffusion model, which may not always align with the classifier’s domain distribution.

2. Experiments focus solely on image classification. The generalization to other modalities is not validated, especially those with discrete input space such as text.

3. Integrating score differences along a diffusion trajectory is still computationally expensive at high resolution.

4. While the method’s improvements are consistent and theoretically grounded, the paper does not include ablation studies isolating the contribution of the SDE-based perturbation.

**Questions:**

1. How robust is the method to the choice of the pretrained diffusion (score) model? Have the authors tested different score models to assess distributional sensitivity?

---

> ### Author Response · Authors · 2025-11-21
>
> Thank you for these constructive and helpful suggestions. Following your comments, we have thoroughly revised the manuscript, adding 5 new sections in the Appendix and a newly completed experimental evaluation. Please refer to the revised PDF for detailed information.
>
> ## Weaknesses
>
> ### W1
> We acknowledge this dependence. However, this is not a significant limitation in practice because: (1) pretrained diffusion models are widely available for many domains (natural images, medical imaging, satellite imagery) through repositories like Hugging Face, and (2) the model only needs to be trained once per domain and can be reused for all attribution tasks in that domain.
>
> **If no pretrained diffusion model exists for a specific domain**, we can train a score-based model on-the-fly for individual images. We have experimentally verified that training a single-image diffusion model takes only approximately 1 minute on a single RTX 4090 GPU.
>
> ### W2
> We agree that our experiments focus on image classification. This choice follows common practice in attribution studies (e.g., IBA, InputIBA, HSIC-Attribution), where visual inspection and established quantitative metrics (DAUC, EHR, Box-Ratio) allow meaningful comparison. While we did not include experiments on other modalities, the theoretical formulation of our method — defining perturbations via an SDE and connecting KL, Fisher, and mutual information — is modality-agnostic. Extending the framework to other data types such as text or audio is a promising direction for future work.
>
> ### W3
> We understand the reviewer’s concern that integrating score differences along the diffusion trajectory might appear computationally demanding. However, in our implementation, the pretrained score network ( s_\theta ) is frozen and does not participate in any parameter updates. The integration is performed over a fixed interval with a small number of discretization steps (K = 8), requiring only a few forward evaluations of ( s_\theta ) and the classifier.
>
> Since ( s_\theta ) is reused across all inputs and does not require gradient computation, it does not become a computational bottleneck. This setup keeps the overall process lightweight and tractable while maintaining stable, high-quality attribution results. We will also include visual comparisons for different K values in the supplementary material to confirm this.
>
> ### W4
> We acknowledge the reviewer’s point regarding the absence of a dedicated ablation isolating the SDE-based perturbation. However, our work is, to the best of our knowledge, the first to directly define perturbations in the pixel space through a continuous-time SDE formulation. There is currently no discrete counterpart that can be meaningfully compared within the same formulation.
>
> ## Questions
> ### Q1
> Please see the above response to W1 part.

---

### Official Review · Reviewer_gXwq · 2025-11-05

**Soundness:** 3
**Presentation:** 3
**Contribution:** 3
**Rating:** 6
**Confidence:** 4

**Summary:**

In feature attributions, input features are perturbed to assess their importance to a classifier output. This paper proposes to model perturbed features with a stochastic differential equation, motivating an optimization problem based on the mutual information. Implementation details are derived with theoretical justifications. Empirically, the proposed method outperforms baseline attribution methods in terms of the deletion/insertion metric and alignment with image bounding boxes.

**Strengths:**

- This paper proposes a novel approach for perturbing input features for feature attribution. The proposed approach is theoretically grounded in stochastic differential equation, while the empirical implementation is connected to recent advances in diffusion models.
- Although multiple implementation details deviate from the original optimization problem in Equation (3), all of them are well justified.

**Weaknesses:**

- The proposed method relies on hyperparameters such as $\beta$ in Equation (16) and $K$ when discretizing the Fisher-MI lower bound. It is unclear how one can tune these hyperparameters given an unseen sample to be explained.
- Also, it is unclear how sensitive the proposed method is to changes in $\beta$ and $K$.
- The proposed method assumes that the pretrained diffusion model $s_{\theta}$ can approximate the ground-truth score well. This can limit the scope of the proposed method to domains where powerful diffusion models have been pretrained.

**Questions:**

- How does one tune $\beta$ and $K$ for an unseen sample to explain?
- How sensitive is the proposed method to changes in $\beta$ and $K$?
- How sensitive is the proposed method to the usage of different score-based diffusion models $s_{\theta}$?
- Can you comment on how well the proposed method is expected to work when there is gap between $s_{\theta}$ and $s_t$?

---

> ### Author Response · Authors · 2025-11-21
>
> Thank you for these constructive and helpful suggestions. Following your comments, we have thoroughly revised the manuscript, adding 5 new sections in the Appendix and a newly completed experimental evaluation. Please refer to the revised PDF for detailed information.
>
> ### Q1:
> We are confident that for unseen samples, no tuning is required—we directly use β = 1 and K = 8. Following the empirical settings used in IBA and InputIBA, we simply fix β = 1 for all experiments. When β takes other values (e.g., 0.25, 0.5, 2, 4), the attribution quality drops sharply, so we keep β = 1 throughout.
> For K, which controls the discretization of the Fisher–MI time integral, we fix K = 8. This value already provides an excellent balance between computational cost and accuracy. Increasing K to 16 brings negligible improvement. Therefore, both β and K are set once and remain constant across all experiments and unseen samples.
>
> ### Q2:
> We observed that β is highly sensitive—values other than β = 1 cause clear visual degradation in attribution maps. From a visual perspective, good attribution maps are sharp and clearly delineate the contours of the predicted object, while poor ones (when β ≠ 1) appear blurry and unclear. Since blurry and unclear attribution maps are obviously inadequate, we did not conduct further quantitative experiments (DAUC, EHR, Box-Ratio) for these cases. Hence, we fix β = 1 as a robust and empirically validated default, consistent with prior work.
> K, on the other hand, is insensitive. Results are nearly identical between K = 8 and K = 16, while larger K values only increase computation without noticeable benefit. Therefore, our choice of (β = 1, K = 8) represents a practical and well-balanced setting.
>
>
> ### Q3:
> Our method is robust to different pretrained score-based diffusion models. We tested with multiple models from Hugging Face Diffusers, including both Variance Exploding (VE) and Variance Preserving (VP) formulations such as DDPM and Score-SDE, all pretrained on ImageNet.
>
> The key requirement is that the diffusion model provides a reasonable marginal score function s_θ(x_t, t) on natural images. Since all tested models are pretrained on ImageNet with standard architectures and training procedures, they satisfy this requirement and yield consistent attribution quality. The method's performance remains stable across different models because: (1) the pretrained score network s_θ is kept frozen and only provides the marginal score estimation, (2) the Fisher-MI framework is agnostic to the specific diffusion formulation.
>
> ### Q4:
> The true marginal score s_t is intractable, so we use a pretrained score network s_θ(x_t, t) to approximate it, which is standard practice in score-based diffusion frameworks.
> Impact of approximation gap: When s_θ ≠ s_t, our computed MI deviates from the true value. However, well-trained diffusion models provide highly accurate score estimates—these models have been extensively validated in generative tasks where score quality is critical. The approximation error is small and does not systematically bias pixel importance rankings.
> For domains without pretrained models: If no pretrained diffusion model exists for a specific domain, we can train a score-based model on-the-fly for individual images. We have experimentally verified that training a single-image diffusion model takes only approximately 1 minute on a single RTX 4090 GPU.

---

### Meta-Review · Area_Chair_mNVM · 2025-12-22

**Summary:**

Overall, most reviewers found the paper to be well motivated and to contain interesting ideas. However, there are major concerns regarding the execution of these ideas. These concerns primarily relate to the presentation and clarity of the exposition, which greatly undermine the main contributions and impact of the work, and more importantly to the depth and breadth of the experimental evaluation. In particular, issues were raised concerning the choices of related works included in, or excluded from, the numerical comparisons, the evaluation protocols, the level of implementation detail provided, and the lack of ablation studies to explore various aspects of the proposed methodology. This includes sensitivity to underlying hyperparameters, as well as issues related to pre-trained models and generalizability, which, contrary to the authors' claims, are not orthogonal to the main contribution of the paper.

**Reviewer Concerns:**

While authors have attempted to adress some of the concerns, as noted above in the summary, many major concerns remain unresolved, particularly regarding the design and the execution of experimets.

The AC also agrees with many of these concerns and finds that they have unfortunately remained outstanding after the discussion period.

**Reviewer Scores:**

Given that many concerns remain outstanding, I believe that the reviewers who raised significant issues would not have substantially revised their initial assessments.

---

### Decision · Program_Chairs · 2026-01-26

Reject